# 🐻 QuoKA: Query-oriented KV selection for Efficient LLM Prefill

**Dalton Jones, Junyoung Park, Matthew Morse, Mingu Lee, Chris Lott, M. Harper Langston**
Qualcomm AI Research *
{daltjone, junpark, mingul, clott, hlangsto}@qti.qualcomm.com

## ABSTRACT

We present QuoKA: Query-oriented KV selection for efficient Attention, a training-free and hardware agnostic sparse attention algorithm for accelerating transformer inference under chunked prefill. While many queries focus on a smaller group of keys in the attention operator, we observe that queries with low cosine similarity with respect to the mean query interact more strongly with more keys and have the greatest contribution to final attention logits. By prioritizing these low cosine similarity queries, the behavior of full attention during the prefill stage can be closely approximated. QuoKA leverages this observation, accelerating attention by (1) first retaining a small set of representative queries and (2) then subselecting the keys most aligned with those queries. Through experiments on Needle-In-A-Haystack, LongBench, RULER, and Math500, we show that, while realizing a $3\times$ reduction in time-to-first-token, $5\times$ speedup in attention on Nvidia GPUs and up to nearly a $7\times$ speedup on Intel Xeon CPUs, QuoKA achieves near-baseline accuracy, utilizing 88% fewer key-value pairs per attention evaluation.

## 1 INTRODUCTION

A major bottleneck in large language model (LLM) inference is prefill latency, which can account for more than 70% of total runtime. This latency is especially significant on CPUs, consumer-grade GPUs, and edge accelerators, where resources are limited (Agrawal et al., 2024; Kamath et al., 2025; Xu et al., 2025a). To mitigate this, recent deployments increasingly adopt *chunked prefill*, which divides input into blocks to improve scheduling and utilization (Agrawal et al., 2023; Lai et al., 2025; Kwon et al., 2023). Nevertheless, due to quadratic complexity of the underlying attention, chunked prefill remains computationally expensive. Methods such as sparse attention seek to overcome this complexity by identifying and exploiting sparsity in attention. Combining chunked prefill with sparse attention offers a promising path to sub-quadratic complexity and substantial latency improvements in resource constrained environments.

Sparse attention algorithms can be broadly categorized into two families: *pattern-based* and *query-dependent* approaches. Pattern-based approaches impose fixed sparsity patterns (e.g., block, strided, or banded) on $QK^\top$ (Child et al., 2019; Dao et al., 2022; Xu et al., 2025b; Jiang et al., 2024; Gao et al., 2024; Zhang et al., 2025), often achieving speedups through kernel-level optimizations; however, due to dynamic compute graph and KV cache memory bandwidth overhead under chunked prefill, their benefits are limited. Furthermore, reliance on custom kernels limits the portability of pattern-based approaches across heterogeneous hardware.

In contrast, query-dependent approaches operate directly on the KV cache, adaptively subselecting the most relevant KVs for a given query (Ribar et al., 2024; Tang et al., 2024; Zhu et al., 2024; Yang et al., 2025b; Singhania et al., 2024). While remaining compatible with optimized kernels and offering strong portability benefits, this strategy reduces both attention complexity and memory traffic. However, existing query-dependent methods are primarily designed for generation, where KVs are selected for a single query; in such cases, it is more straightforward to determine which KVs are relevant for the given query. During prefill, when relevant KVs are selected for many queries at once, this can result in significant performance degradations. Under chunked prefill, where important KVs are repeatedly subselected for multiple queries, these degradations become more pronounced.

---

*Qualcomm AI Research is an initiative of Qualcomm Technologies, Inc.

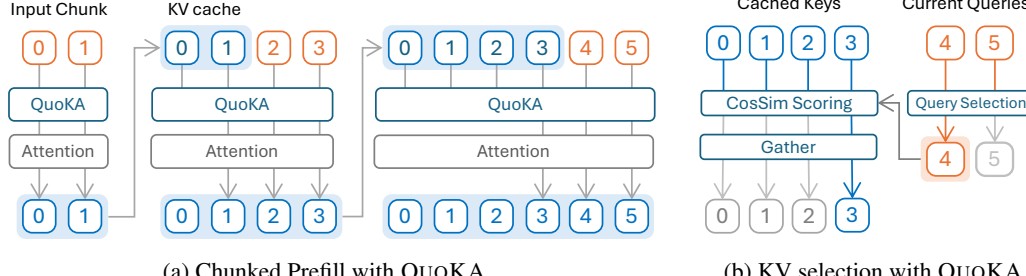

(a) Chunked Prefill with QuoKA  (b) KV selection with QuoKA

Figure 1: **Overview of chunked prefill with QUOKA: (a)** A prompt of 6 tokens is divided into three chunks of 2 tokens each. For each chunk, QUOKA subselects the KV cache and feeds the reduced cache into a dense attention kernel. **(b)** Subselection is performed by applying query subselection based on cosine dissimilarity, followed by key subselection using query–key cosine similarity.

To address the drawbacks of existing approaches, we propose QUOKA: Query-oriented $\underline{K}$V selection for efficient $\underline{A}$ttention, a *training-free sparse attention method optimized for chunked prefill* built upon the following observation: **queries with lower cosine similarity to the mean query attend to the majority of keys**.

QUOKA leverages this observation to retain a small subset of representative queries and subselect KVs with which they strongly interact. While preserving accuracy, QUOKA accelerates attention during prefill and reduces the number of KVs through three steps: *Query Subselection*, which retains only the most informative queries; *Cosine-Similarity Scoring*, which provides a stable, bounded proxy for query-key relevance; and *Group-Aware Aggregation*, which efficiently preserves compatibility with modern architectures such as grouped-query attention (GQA) (Ainslie et al., 2023). These steps can be seen in Figure 1, discussed in greater detail in Section 3. Unlike kernel-level sparsity, which depends on custom primitives, QUOKA relies on standard linear algebra operations, allowing for platform portability and straightforward deployment. Our contributions are summarized as follows:

- **QUOKA**: a hardware-agnostic and training-free query-oriented sparse attention method for chunked prefill, built on standard linear algebra kernels.
- **Accuracy under sparsity:** near-baseline results on long-context benchmarks (Needle-in-a-Haystack, RULER, LongBench, Math500), outperforming existing sparse attention methods.
- **Latency reduction:** up to $5\times$ attention speedup and $3\times$ lower time-to-first-token (TTFT) on enterprise-class GPUs, and $7\times$ and $5\text{-}6\times$ speedups on CPUs and on consumer GPUs.
- **Generalization across architectures:** validated on diverse decoder-only LLM families (Llama3, Qwen3, SmolLM, GPT-OSS) and on RoPE/NoPE and MoE-based LLMs.
- **Robustness to hyperparameters:** accuracy degrades gradually with sparsity and remains stable across parameter choices, enabling deployment under varied constraints.

## 2 BACKGROUND

### 2.1 TRANSFORMERS AND ATTENTION

Since their introduction by Vaswani et al. (2017), transformers have become the dominant sequence modeling paradigm. A transformer block consists of two main computational steps: the attention operator and a feed-forward network (FFN). In this work, we focus on the attention primarily.

For an input sequence $X = [x_1, \ldots, x_T]$, the attention operator $\text{Attn}(X)$ is defined as

$$\text{Attn}(X) = \text{Softmax}\left(QK^\top/\sqrt{d} + M\right)V = AV, \tag{1}$$

where keys $K = XW_K$, queries $Q = XW_Q$, and values $V = XW_V$ are learned projections of the input tokens, with $W_K, W_Q, W_V \in \mathbb{R}^{d \times d}$. With zeros below the diagonal and $-\infty$ elsewhere, the mask $M$ enforces autoregressive causality.

---

**Algorithm 1** KV cache sub-selection using QUOKA

---

**Require:** queries $Q$, keys $K$, values $V$, prefill chunk size $B_{\text{CP}}$, selective attention budget $B_{\text{SA}}$, max queries $N_Q$, number of KV heads $n_{\text{KV}}$, number of attention heads $n_Q$

$\triangleright$ Query Sub-selection

1: **if** $B_{\text{CP}} > N_Q$ **then**
2:      $M_Q \leftarrow \texttt{mean}(Q, \texttt{dim=2})$
3:      $S_Q \leftarrow \text{CosSim}(Q, M_Q)$
4:      $Q \leftarrow \texttt{gather}(\texttt{topk}(-S_Q, N_Q), Q)$
5: **end if**

$\triangleright$ Efficient Cosine Similarity via Pre-aggregation

6: $Q \leftarrow Q/\texttt{norm}(Q, \texttt{dim=-1})$          $\triangleright (b, n_q, N_Q, d)$
7: $K \leftarrow K/\texttt{norm}(K, \texttt{dim=-1})$          $\triangleright (b, n_{\text{KV}}, T, d)$
8: $\bar{Q} \leftarrow \texttt{mean}\Big(Q.\texttt{reshape}(b, n_{\text{KV}}, \frac{n_Q}{n_{\text{KV}}}, N_Q, d), \texttt{dim=2}\Big)$      $\triangleright (b, n_{KV}, N_Q, d)$
9: $S \leftarrow \bar{Q} K^{\top}$          $\triangleright (b, n_{\text{KV}}, N_Q, T)$
10: $\hat{S} \leftarrow \texttt{max}(S, \texttt{dim=2})$          $\triangleright (b, n_{\text{KV}}, T)$
11: $I \leftarrow \texttt{topk}(\hat{S}, B_{\text{SA}})$
12: $K^{\star}, V^{\star} \leftarrow \texttt{gather}(K, I), \texttt{gather}(V, I)$

---

## 2.2 ATTENTION LATENCY: PREFILL VS GENERATION

Autoregressive text generation in transformer LLMs can be divided into two stages: *prefill* and *generation*, both with distinct latency characteristics. During the *prefill* stage, the entire input prompt is processed to initialize the KV cache, requiring $O(T^2)$ operations from dot products between $T$ queries and $T$ keys in addition to the FFN computation for each token. As a result for long prompts, attention dominates prefill latency. Optimized kernels such as FlashAttention (Dao et al., 2022; Dao, 2023; Shah et al., 2024) improve memory locality and utilization, but they do not reduce the quadratic complexity; asymptotic speedups can be achieved by reducing the effective number of KVs attended to by each query. In the generation stage, a single new query attends to the $T$ cached KVs. For short generation lengths, performance is memory-bound by the large transfer cost of FFN weights. However for longer outputs, the KV cache dominates the memory footprint and bandwidth. Reducing the number of stored KVs both speeds up the attention computation and reduces data transfer, making KV reduction essential for accelerating generation-heavy tasks (e.g., reasoning or code generation).

## 2.3 CHUNKED PREFILL

Given the causality in Equation (1), *chunked prefill* (Agrawal et al., 2023; Holmes et al., 2024) partitions the input sequence $X$ into non-overlapping chunks of size $B_{\text{CP}}$ and process the chunks sequentially: $X = [X_0, X_1, \ldots, X_{N_B-1}]$, $N_B = \lceil T/B_{\text{CP}} \rceil$, where $X_i = [x_{B_{\text{CP}}i}, \ldots, x_{B_{\text{CP}}(i+1)-1}]$ denotes the $i$-th chunk. For chunk $i$, let $K_{<i}$ and $V_{<i}$ denote the concatenation of keys and values from all preceding chunks. Then:

$$\text{Attn}(X_i) = \text{Softmax}\Big(Q_i [\, K_i \mid K_{<i} \,]^{\top}/\sqrt{d} + M_i\Big) [\, V_i \mid V_{<i} \,], \qquad (2)$$

where $[\,\cdot \mid \cdot\,]$ denotes concatenation along the sequence dimension and $M_i$ enforces causality within $X_i$ and against future tokens. Chunked prefill is particularly important for inference on edge devices such as mobile hardware or consumer GPUs where bandwidth and memory capacity are constrained. It also improves cloud inference throughput by enabling interleaved prefill and decode requests, increasing GPU utilization (Holmes et al., 2024; Agrawal et al., 2025).

## 2.4 ATTENTION SPARSITY AND SPARSE ATTENTION

High sparsity in the attention matrix $A$ has been widely observed in LLMs (Zhang et al., 2024; Xiao et al., 2024; Oren et al., 2024; Zhu et al., 2024; Park et al., 2025). This sparsity allows for a reduction in attention cost by selecting the most relevant KVs for incoming queries into an active cache $(\hat{K}, \hat{V})$ such that:

$$I = \texttt{topk}(f(Q, K), B_{SA}), \quad \hat{K} = \texttt{gather}(K, I), \quad \hat{V} = \texttt{gather}(V, I), \qquad (3)$$

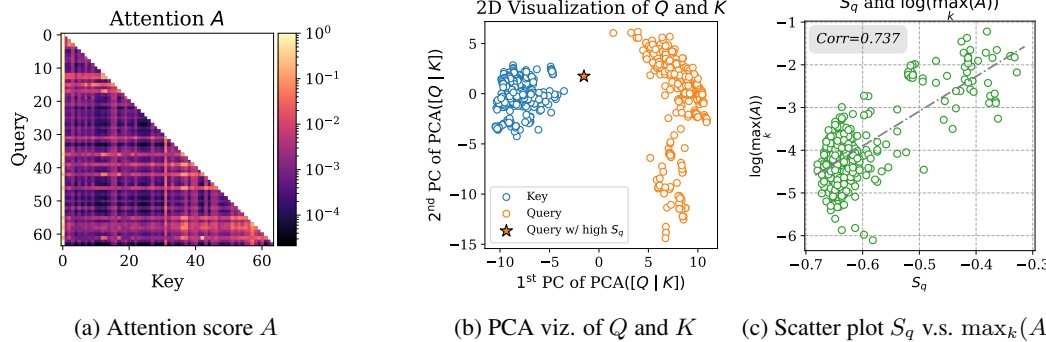

Figure 2: **Empirical observations from *Llama 3.2-3B-Instruct*, layer 0 head 11.** (a) Attention map $A$. (b) PCA visualization of $Q$ and $K$, showing that queries with higher $S_q$ lie closer to the keys. (c) Correlation between $S_q$ and $\max_k(A)$, indicating stronger key interactions for higher $S_q$ queries.

where $f$ satisfies

$$\underset{f(Q,K)}{\text{minimize}} \left\| \text{Softmax}\left(QK^\top/\sqrt{d} + M\right)V - \text{Softmax}\left(Q\hat{K}^\top/\sqrt{d} + M\right)\hat{V} \right\|, \quad (4)$$

where $f$ is usually constrained to be more efficient than the attention computation itself.

Prior work (Zhao et al., 2019; Ribar et al., 2024; Tang et al., 2024; Singhania et al., 2024; Yang et al., 2024b) generally focused on the generation phase, where the scoring function $f$ operates on a single query $Q$. Extending these methods to the multiple-query setting with by averaging over queries significantly degrades performance (see Table 3), highlighting the need for better ways to aggregate information from multiple queries. While recent work Zhang et al. (2025); Gao et al. (2024); Zhu et al. (2024); Jiang et al. (2024); Lai et al. (2025) attempt to address this, these methods typically depend on custom CUDA kernels on NVIDIA GPUs. This limits compatibility with hardware-tuned kernels like FlashAttention and hinders broader deployment outside of data centers.

## 3 QUOKA METHOD

As discussed in Section 2, existing sparse attention methods face limitations in prefill efficiency and portability. QUOKA addresses this by reducing the active KV cache during the chunked prefill process. Specifically, the input sequence $X$ is divided into chunks $\{X_0, X_1, \ldots, X_{N_B-1}\}$, and for each chunk $X_i$, QUOKA selects only the most relevant KVs before computing attention as shown in Figure 1. This is achieved in three stages, as detailed in Algorithm 1.

1. **Query subselection**: retain only the most informative queries to reduce redundancy.
2. **Cosine-Similarity Scoring**: compute cosine similarity between queries and keys to estimate their relevance.
3. **Score Aggregation**: combine scores across queries and key-value groups to select the final KV.

The reduced KV set for $X_i$ is then fed into a dense attention kernel such as FlashAttention. By applying this repeatedly, QUOKA reduces prefill cost from $O(T^2)$ to a sub-quadratic complexity.

### 3.1 QUERY SUBSELECTION

Prior work (Park et al., 2025) examined the distinctive geometric characteristic of keys in LLM. Inspired by this insight, we focus on the geometry of queries. We observe that queries with lower cosine similarity to the mean query tend to align broadly with most keys, while near-mean queries concentrate on a small shared group of keys (Figure 2a). Thus, utilizing all queries yields redundant information and increases complexity.

To mitigate this, QUOKA retains only those queries that consistently exert significant influence on keys. These queries can be identified using their angular distance from the average query vector $M_Q$. Formally, we rank each query $q$ by $-\text{CosSim}(M_Q, q)$ and retain the top $N_Q$. This preserves

the queries most responsible for large key-query dot products while normalizing per query, allowing us to efficiently approximate the post-softmax attention matrix $A$. This can be formalized through the following theorem:

**Theorem 1.** *Consider tokens a fixed query $q_0$ and key $k$, and let the average of a set of queries be denoted $M_Q$. Suppose $CosSim(k, q_0) = \beta_q > 0$ and $CosSim(M_Q, k) = \alpha_q < 0$. Then*

$$CosSim(M_Q, q^*) \leq 1 + \alpha_q \beta_q - 0.5\alpha_q^2 - 0.5\beta_q^2. \tag{5}$$

The proof is provided in Appendix D. Intuitively, if a query $q$ attends strongly to $k$, then $\beta_q$ will be large while $\alpha_q$ will be small, resulting in a large subselection score

$$S_q = -\text{CosSim}(M_Q, q^*).$$

Hence our subselection retains queries that contribute most to the attention distribution, in line with both empirical observations and the underlying geometry of keys and queries in modern LLMs. To further validate our design choice, we analyze the geometry of queries and keys. As shown in Figure 2b, most queries are separated from the key cluster in a 2D PCA projection, while queries with higher $S_q$ tend to lie closer to the keys. Complementing this, Figure 2c demonstrates that higher $S_q$ is correlated with larger $\max_k(A)$ outside of the sink token, indicating that such queries exert stronger influence on individual keys. Together, these observations suggest that $S_q$ identifies queries that are both geometrically aligned with keys and dominant in attention.

## 3.2 SCORING VIA COSINE SIMILARITY

Given the reduced set of queries, the next step is to evaluate their interactions with keys. Existing methods often use dot products $QK^\top$ but these are scale-dependent and unstable under aggregation. Instead, QUOKA computes

$$S = \text{CosSim}(Q, K),$$

which normalizes vectors to unit length and provides a bounded, geometry-aware proxy for softmax attention weights. Recent work (Mongaras et al., 2025; Park et al., 2025) also supports the use of cosine similarity as a lightweight normalization that approximates softmax behavior. Empirically, ablations on the RULER benchmark in Table 9 show that cosine similarity improves subselection quality by more than 10% compared to the dot product.

## 3.3 AGGREGATION ACROSS QUERIES AND KV GROUPS

Aggregation of scores is required along two axes: across queries and across grouped-query attention (GQA) heads. For the query axis, averaging can obscure rare but important query–key interactions, so QUOKA instead takes the **maximum**, which preserves such outliers. This is supported by the heavy-tailed distribution in Figure 3 and by gains on the RULER benchmark (Table 10). For the GQA axis, we simply average scores across KV groups. Unlike queries, head-level importance is correlated (Bhojanapalli et al., 2021), making the **mean** accurate and stable. Notably, if we normalize $K$ and $Q$ prior to computing the score, we can achieve the same average by *pre-aggregation*: averaging normalized queries across KV groups due to the

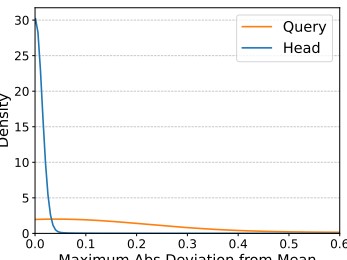

Figure 3: Distribution of attention score max deviation from mean along query and head dimension.

linearity of the mean and the outer product $QK^T$. Pre-aggregation also lowers computation and memory cost by a factor of the number of KV groups (which is large in most modern models), enabling compatibility with modern architectures and enhancing efficiency.

## 3.4 CHUNKED PREFILL WITH QUOKA

The three components described above—query subselection, scoring, and aggregation—are integrated into the overall *chunked prefill* process. For each incoming block $X_i$, QUOKA subselects the active KV tokens using the procedure in Algorithm 1. The resulting subset of keys and values is then passed to the attention computation for that chunk. By reducing the KV budget at each chunk, QUOKA shrinks both compute and memory transfer costs. This enables significant prefill acceleration while maintaining accuracy across long-context benchmarks and is summarized in Algorithm 2.

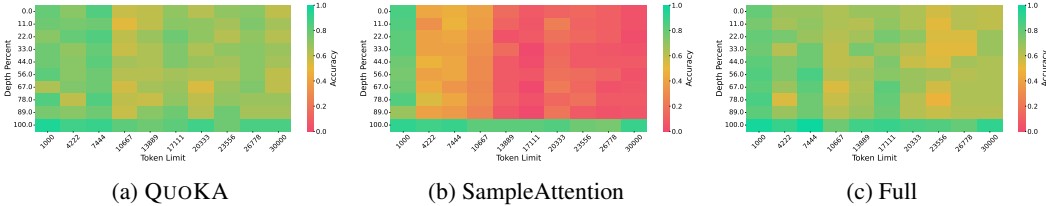

(a) QUOKA        (b) SampleAttention        (c) Full

Figure 4: Accuracy across document length and needle depth for NIAH with $B_{\text{SA}} = 2048$ and $B_{\text{CP}} = 128$. Results for additional sparse attention methods are shown in Figure 7.

## 4 RESULTS

In this section, we empirically demonstrate the effectiveness of QUOKA. We begin with a detailed description of the evaluation setup and sparse attention baselines, then present our results.

**Sparse Attention Baselines:** We compared a group of sparse attention algorithms on a set of models across a variety of selective budgets. The competing algorithms include SAMPLEATTENTION (Zhu et al., 2024) which uniformly samples queries to compute $S$, LESSISMORE (Yang et al., 2025b) which computes $S$ only at specified layers, SPARQ (Ribar et al., 2024) which subselects along channel dimension before computing $S$, and LOKI (Singhania et al., 2024) which down-projects keys and queries before computing $S$. Unless otherwise specified, QUOKA and SAMPLEATTENTION subselect 16 queries at a time while SPARQ and LOKI down project keys and queries to 64 dimensions.

**Models and Evaluation Setup:** To evaluate the efficacy of QUOKA across diverse model structures, we adopted QUOKA and baseline methods across a variety of model families, including simple attention-only models, MoE-based FFN variants, and NoPE adaptations. The models considered include Llama 3.2-3B-Instruct (Dubey et al., 2024), Qwen 2.5-3B-Instruct (Yang et al., 2024a), Smollm3 (Bakouch et al., 2025), Qwen3-4B, Qwen3-30B-A3B (Yang et al., 2025a), and GPT-OSS-20B (Agarwal et al., 2025). Additionally, we simulated a resource constrained scenario by utilizing chunked prefill with a block size $B_{\text{CP}} = 128$. Unless otherwise specified, experiments were performed utilizing an Nvidia A100 GPU.

**Evaluation Summary:** We show that QUOKA successfully preserves downstream accuracy on benchmarks involving long input prompts, extends naturally to generation-intensive tasks such as mathematical reasoning, and provides both attention module-level and end-to-end speedup. Results are summarized below:

- **Needle-In-a-Haystack.** QUOKA significantly outperforms competing selective attention methods on the needle-in-a-haystack (NIAH) benchmark (Section 4.1).

- **RULER.** QUOKA significantly outperforms competing methods on the RULER benchmark Hsieh et al. (2024), achieving consistently better performance across all prompt lengths and models (Section 4.2).

- **LongBench.** QUOKA exceeds the performance of competing baselines on the LongBench suite Bai et al. (2024), demonstrating substantially less performance degradation with decreasing selective budgets. Across all models and budgets, chunked prefill with QUOKA outperforms other methods by at least 10–20% (Section 4.3).

- **Math500.** Although our primary focus is prefill, QUOKA is also directly applicable to generation. On the Math500 benchmark, QUOKA outperforms a sparse attention method specialized for generation and in some cases surpasses the accuracy of dense attention (Section 4.4).

- **Ablation.** In sweeping over combinations of $B_{\text{CP}}$, $B_{\text{SA}}$, and $N_Q$ that govern the efficiency–accuracy trade-off in QUOKA, we observe that accuracy decreases only gradually as sparsity increases. This indicates that QUOKA can be tuned for diverse hardware environments while maintaining high efficiency. (Section 4.5)

- **Latency.** We measured time-to-first-token (TTFT) across increasing prompt lengths for end-to-end inference, along with standalone attention module latency. QUOKA achieves a $5\times$ module-level speedup on 30k tokens and a $3\times$ TTFT improvement on 50k tokens relative to the base model with full attention, while scaling more efficiently than competing sparse methods (Section 4.6).

Table 1: **RULER evaluation results across increasing lengths with** $B_{\text{SA}} = 1,024$. Highest per column in **bold** (Higher is better). Attention sparsification applied on full attention layers.

| | Llama-3.2-3B-Instruct | | | | Qwen-2.5-3B-Instruct | | | | Qwen-3-4B | | | | Smollm3 | | | | GPT-OSS-20B | | | |
|---|---|---|---|---|---|---|---|---|---|---|---|---|---|---|---|---|---|---|---|---|
| Length | 4k | 8k | 16k | 32k | 4k | 8k | 16k | 32k | 4k | 8k | 16k | 32k | 4k | 8k | 16k | 32k | 4k | 8k | 16k | 32k |
| SnapKV | 29.15 | 13.70 | 12.44 | 6.21 | 19.25 | 12.40 | 9.88 | 8.13 | 27.29 | 17.15 | 10.94 | 10.07 | 43.72 | 35.98 | 24.21 | 12.23 | 58.18 | 31.45 | 24.82 | 16.63 |
| KeyDiff | 53.34 | 31.10 | 24.29 | 14.87 | 34.09 | 20.68 | 15.79 | 10.32 | 37.29 | 27.15 | 20.94 | 12.07 | 62.85 | 51.60 | 41.64 | 30.37 | 69.58 | 28.76 | 15.86 | 9.65 |
| LessIsMore | 75.15 | 49.23 | 30.44 | 19.16 | 36.66 | 20.21 | 12.84 | 10.12 | 65.55 | 42.75 | 24.39 | 14.87 | 79.67 | 50.17 | 35.12 | 24.21 | 67.35 | 54.49 | 38.27 | 20.11 |
| Loki | 74.84 | 56.50 | 25.76 | 8.05 | 74.40 | 60.09 | 48.96 | 34.12 | 82.83 | 65.19 | 52.29 | 39.31 | 84.52 | 64.20 | 50.10 | 22.66 | 75.48 | 65.36 | 54.67 | 39.92 |
| SparQ | 79.36 | 60.80 | 48.59 | 31.14 | 78.07 | 59.87 | 54.71 | 36.74 | 87.93 | 68.97 | 56.02 | 35.20 | 82.45 | 57.62 | 32.18 | 18.69 | 70.07 | 54.00 | 30.75 | 15.20 |
| SampleAttn | 78.25 | 61.14 | 48.31 | 31.73 | 77.17 | 60.88 | 56.64 | 36.17 | 87.84 | 72.46 | 59.57 | 40.72 | 85.72 | 66.44 | 59.10 | 45.98 | 76.20 | 70.35 | 53.91 | 30.42 |
| QUOKA | **86.71** | **80.19** | **70.90** | **57.01** | **87.85** | **74.27** | **66.82** | **59.37** | **93.73** | **91.07** | **88.57** | **74.83** | **89.97** | **79.94** | **72.69** | **61.37** | **78.92** | **79.19** | **73.40** | **57.79** |

We provide additional runtime and memory complexity analysis for QUOKA and the baselines in Appendix C.

## 4.1 NEEDLE IN A HAYSTACK

NIAH is a synthetic benchmark designed to evaluate how well large language models retrieve specific information from long contexts (Kamradt, 2023; Liu et al., 2024). A single *needle* sentence containing a key fact is inserted into a large body of irrelevant text (i.e., *haystack*), and the model is asked a question that can only be answered by finding that needle. We test Llama3.2-3B-Instruct with selective budget $B_{\text{SA}} = 2048$ and sequence lengths up to 30K. Figure 4 shows that most selective attention methods incur significant degradation during chunked prefill. However, QUOKA retains the ability to retrieve important information across various input sequence lengths.

## 4.2 RULER BENCHMARK

RULER (Hsieh et al., 2024) is a synthetic test for evaluating long-context capabilities of LLMs using NIAH and QA tasks. It addresses shortcomings of NIAH by extending beyond basic retrieval to include multiple needles and new categories such as multi-hop tracing and aggregation, which require reasoning across dispersed information rather than locating a single token.

We ran experiments to evaluate QUOKA against sparse attention methods and the full attention baseline. Table 1 reports results with $B_{\text{SA}} = 1024$, showing 10–20% higher scores than baselines. We also simulated $B_{\text{SA}}$ growing with the KV cache to maintain a constant compression ratio: during chunked prefill and token generation, $B_{\text{SA}}$ is set to 25% of full cache length. Table 2 shows that accuracy loss remains very limited even at long sequences.

Table 2: **RULER eval of QUOKA** with $B_{\text{SA}}$ set to 25% of the KV cache length

| Model | Budget | Prompt Length | | | |
|---|---|---|---|---|---|
| | | 4096 | 8192 | 16384 | 32768 |
| Llama3.2-3B | Full | 87.50 | 81.33 | 78.98 | 76.31 |
| | 25% | 86.94 | 79.72 | 76.02 | 74.14 |
| Qwen2.5-3B | Full | 89.56 | 81.99 | 76.09 | 71.69 |
| | 25% | 86.07 | 79.78 | 74.25 | 68.84 |
| Qwen3-4B | Full | 93.32 | 91.68 | 91.18 | 88.54 |
| | 25% | 92.50 | 91.35 | 90.63 | 87.87 |
| Qwen3-30B A3B-Instruct | Full | 94.08 | 93.75 | 92.02 | 91.87 |
| | 25% | 93.25 | 92.77 | 91.90 | 91.08 |
| Smollm3 | Full | 91.12 | 83.46 | 80.11 | 75.18 |
| | 25% | 89.60 | 81.45 | 78.72 | 73.55 |
| GPT-OSS-20B | Full | 79.35 | 79.32 | 79.78 | 75.47 |
| | 25% | 77.40 | 80.66 | 77.17 | 73.45 |

## 4.3 LONGBENCH

LongBench (Bai et al., 2024) is a multi-task benchmark designed to evaluate the ability of LLMs in handling long sequences. It includes real-world tasks with inputs such as books, technical reports, and multi-document collections. The benchmark features substantial input lengths average 12K tokens and maximum exceeding 60K tokens measured with the Llama tokenizer, making it a rigorous test for models that aim to process extended inputs effectively.

Table 3 reports the normalized accuracy of sparse attention methods across various models and values of $B_{\text{SA}}$. Each cell shows relative scores compared to the dense baseline (where 1.0 indicates no accuracy drop). Across models, we observe that QUOKA maintains minimal accuracy degradation even with a small budget (e.g., $B_{\text{SA}} = 1,024$). Moreover, QUOKA consistently outperforms competing methods across both models and budget settings.

## 4.4 MATH 500 BENCHMARK

Although our primary focus is prefill optimization, QUOKA is also directly applicable to the generation phase, where the model computes attention for a single query and no query subselection is

Table 3: **LongBench results (Higher is better)**. For each model, the three columns correspond to selective budgets $B_{SA} \in \{512, 1024, 2048\}$.

| | Llama3.2-3B-Instruct | | | Qwen2.5-3B-Instruct | | | Qwen3-4B | | | Smollm3 | | |
|---|---|---|---|---|---|---|---|---|---|---|---|---|
| Budget | 512 | 1024 | 2048 | 512 | 1024 | 2048 | 512 | 1024 | 2048 | 512 | 1024 | 2048 |
| LessIsMore | 0.703 | 0.788 | 0.850 | 0.461 | 0.556 | 0.659 | 0.665 | 0.773 | 0.868 | 0.765 | 0.842 | 0.918 |
| SparQ | 0.721 | 0.802 | 0.842 | 0.636 | 0.726 | 0.808 | 0.686 | 0.782 | 0.872 | 0.725 | 0.805 | 0.922 |
| Loki | 0.686 | 0.757 | 0.842 | 0.589 | 0.671 | 0.787 | 0.622 | 0.782 | 0.872 | 0.384 | 0.801 | 0.622 |
| SampleAttn | 0.738 | 0.800 | 0.901 | 0.660 | 0.756 | 0.828 | 0.755 | 0.875 | 0.947 | 0.856 | 0.929 | 0.966 |
| QUOKA | **0.945** | **0.972** | **0.986** | **0.869** | **0.945** | **0.977** | **0.966** | **0.992** | **0.995** | **0.998** | **1.03** | **1.028** |

applied. To demonstrate its efficacy in this setting, we evaluated QUOKA on a generation-intensive reasoning task, applying chunked prefill whenever applicable and using sparse attention for the computation. As shown in Table 8, QUOKA outperforms a sparse attention method specifically designed for generation, and in some cases even surpasses the accuracy of dense attention.

## 4.5 ABLATION STUDY

To examine the efficiency–accuracy trade-off of QUOKA, we performed ablations over the key $B_{SA}$, $B_{CP}$ and $N_q$ parameters . Tables 3, 5 and 6 for both LongBench and Ruler varying $B_{SA}$ demonstrate that across models and benchmarks, accuracy decreases very gradually as sparsity increases while latency and memory usage improve substantially. We achieve less than a 3% drop in performance with less than 12% of the original tokens used for attention. As shown in Table 11, QUOKA maintains stable performance under varying $B_{CP}$. In Table 12 we observe that even with a small setting of $N_q = \frac{1}{16} B_{CP}$, accuracy drops by only ~3% compared to full attention. This robustness allows practitioners to tune QUOKA for diverse hardware constraints with minimal quality loss.

## 4.6 ATTENTION LATENCY AND TTFT

To evaluate the efficiency of QUOKA, we measured latency in two settings: standalone attention modules and time-to-first-token (TTFT) on the Qwen3-4B model. All experiments use the HuggingFace implementation with `bfloat16` precision and FlashAttention for the dense baseline. Each data point is averaged over 100 trials, and hyperparameters follow original publications. To compare QUOKA with existing sparse attention methods, we measured performance across increasing input sequence lengths. All reported times are expressed as speedups relative to the full attention module, and in the end-to-end case, the base Qwen3-4B model with chunked prefill ($B_{CP} = 128$). Figures 5a and 5b show QUOKA scales substantially better than the dense baseline and consistently outperforms or matches the strongest sparse attention competitors on Nvidia A100. QUOKA also achieves significant speedups on consumer-grade hardware: Figures 5c and 5d show 5–6× gains at long context lengths on both an Intel Xeon W-2125 CPU and an Nvidia RTX 2080 GPU, and in decoding (Figure 6).

## 5 RELATED WORK

As discussed in Section 2, the computational bottleneck in prefill stems from the quadratic growth of attention with the size of the KV cache. Under chunked prefill, this cost can be mitigated by subselecting the KV pairs used in the attention. In this context, we review three related lines of work.

**Dynamic query-dependent sparse attention:** Query-dependent sparse attention methods select a subset of cached keys for each query using lightweight proxies for attention scores (Singhania et al., 2024; Tang et al., 2024; Yang et al., 2024b; Ribar et al., 2024). While effective for single-query decode, naïvely averaging such proxies across multiple queries in a prefill chunk often degrades accuracy because these methods are tuned for generation-time settings. SampleAttention (Zhu et al., 2024) targets prefill but treats multiple queries homogeneously. Since queries can exhibit distinct geometry with respective to keys, such a geometry-aware query and key subselection is desirable. QUOKA first selects representative queries via cosine dissimilarity and then selects keys via cosine similarity, enabling higher sparsity at comparable accuracy.

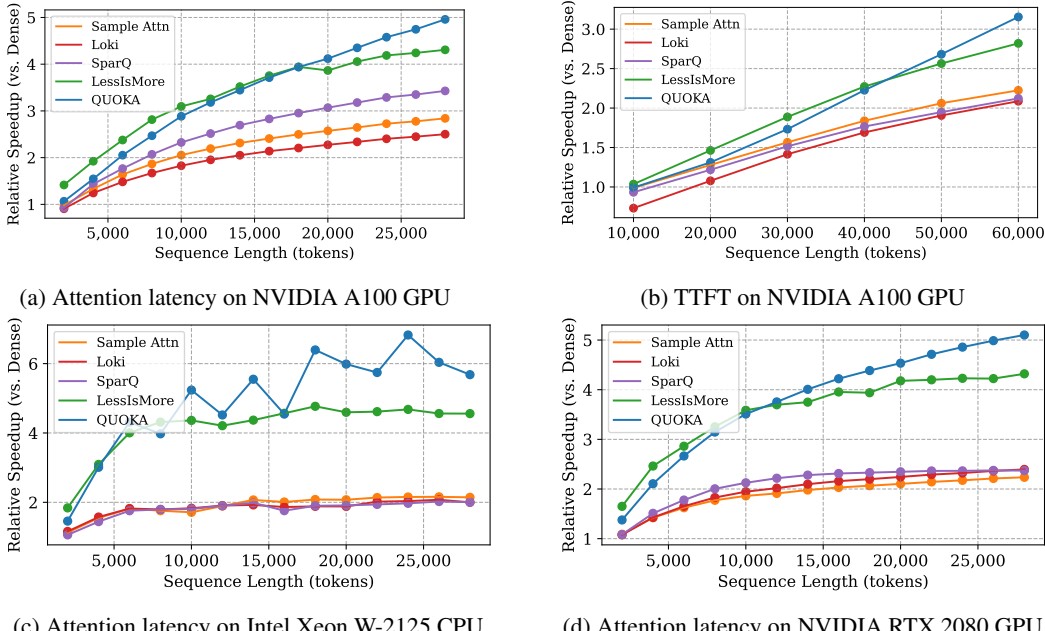

Figure 5: Relative speedup of attention and TTFT compared to dense attention baseline using $B_{\text{CP}} = 128$ on different hardware.

**KV cache eviction:** KV cache eviction removes low-salience KV pairs to reduce memory footprint based on pre-defined patterns, attention scores, or cosine similarity of the keys (Xiao et al., 2024; Zhang et al., 2024; Li et al., 2024; Oren et al., 2024; Park et al., 2025). Like query-dependent sparsification, most eviction policies are designed for single-query generation; a few works aggregate importance across multiple queries (Li et al., 2024; Kim et al., 2024), but typically treat queries homogeneously and remain generation-centric. Eviction is complementary to our approach and could be integrated with QUOKA in chunked prefill; we leave this to future work.

**Kernel-level sparse attention:** Another line of work accelerates prefill through kernel-level sparse attention, typically relying on predefined patterns such as block, band, or strided sparsity (Child et al., 2019; Dao et al., 2022; Xu et al., 2025b; Jiang et al., 2024; Lai et al., 2025). While effective with specialized CUDA implementations, these methods require hardware-specific kernel optimizations and often incur additional overhead. In particular, under chunked prefill the repeated kernel invocations and memory transfers substantially reduce their efficiency. By contrast, QUOKA is fully compatible with standard dense kernels and avoids such hardware and runtime dependencies.

## 6 Conclusion

We introduced QUOKA, a training-free and hardware-agnostic sparse attention mechanism that reduces LLM inference latency. Leveraging the geometry of queries and keys, QUOKA identifies and retains the most influential queries to score and subselect the KV cache for attention without sacrificing accuracy. Across LongBench and RULER, QUOKA achieves near-dense performance with a fraction of the budget while outperforming competing sparse attention methods. On Math500 QUOKA further demonstrates versatility by surpassing generation-specific methods and even dense attention in some cases. In addition, QUOKA provides substantial efficiency gains, including up to $5\times$ standalone attention speedup and $3\times$ TTFT reduction across different devices, suggesting that selective attention is a promising direction for TTFT and TPS optimization. An avenue for future improvement is to make the computation of $\bar{Q}K^\top$ more efficient, for example by exploiting query/key channel sparsity or learned low-dimensional projections.

## 7 ETHICS STATEMENT

This work presents QUOKA, a hardware-agnostic sparse attention algorithm designed to accelerate transformer inference. Our research does not involve human subjects, sensitive personal data, or datasets that raise privacy or security concerns. All datasets used (Needle-In-A-Haystack, LongBench, RULER, and Math500) are publicly available and widely used in the research community. To our knowledge, QUOKA does not introduce methodologies or applications that pose foreseeable risks of misuse or harm. We have no conflicts of interest or sponsorship to declare. We believe this work adheres to the code of ethics and does not raise any ethical concerns.

## 8 REPRODUCIBILITY STATEMENT

We have taken care to ensure that QUOKA is readily reproducible. The main paper includes a detailed description of the algorithm, including its design principles, KV scoring/subselection strategy, and implementation using standard linear algebra routines. We provide comprehensive experimental setups, including datasets used, evaluation metrics, and hardware specifications. All hyperparameters and test configurations are documented in the main text and appendix. While we have not included source code, the simplicity of QUOKA's training-free design makes it straightforward to implement using widely available libraries.

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

# QUOKA
# Supplementary Material

## A    CHUNKED PREFILL WITH QUOKA

We provide the full algorithm of QUOKA used in chunked prefill. Algorithm 2 summarizes the optimized prefill processes with QUOKA under chunked prefill.

---
**Algorithm 2** Chunked Prefill with QUOKA

---
**Require:** Input sequence $X = [X_0, X_1, \ldots, X_{B-1}]$, chunk size $L$, KV budget $B_{\text{SA}}$
1: Initialize output list $\mathcal{Y} \leftarrow [\ ]$
2: **for** each chunk $X_i$ **do**
3:  Compute queries $Q_i$, keys $K_i$, values $V_i$ from $X_i$
4:  $(\hat{K}, \hat{V}) \leftarrow \text{QUOKA}(Q_i, K_{<i}, V_{<i}, B_{\text{SA}})$          ▷ Subselect KV cache
5:  $Y_i \leftarrow \text{ATTENTION}(Q_i, [\hat{K} \mid K_i], [\hat{V} \mid V_i])$
6:  Append $Y_i$ to $\mathcal{Y}$
7:  $K_{<i+1} \leftarrow [K_{<i}|K_i], \ V_{<i+1} \leftarrow [V_{<i}|V_i]$           ▷ Update KV cache
8: **end for**
9: **return** $Y \leftarrow \text{Concat}(\mathcal{Y})$

---

## B    ADDITIONAL TIMING RESULTS

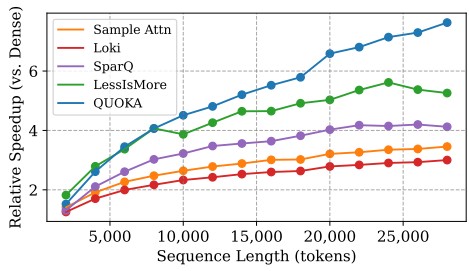 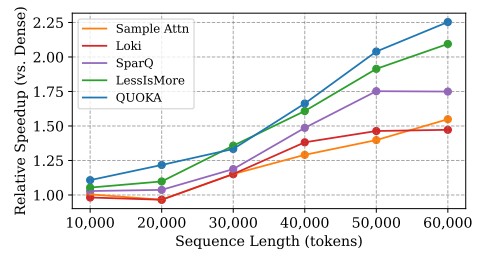

(a) Attention latency (decode) on NVIDIA A100   (b) End-to-End latency (decode) on NVIDIA A100

Figure 6: Speedup versus full attention of an increasing number of decoding steps for a standalone attention module and end to end model (Qwen3-4B). Timings were measured on an Nvidia A100 GPU across 100 trials.

## C    RUNTIME AND MEMORY COMPLEXITY

We compute the runtime and memory complexity for QUOKA and competing sparse attention methods in Table 4. For prefill chunk size $B_{\text{CP}}$, KV cache length $T$, number of $KV$ heads $n_{\text{KV}}$, number of heads $n_{\text{Q}}$, hidden dimension $d$ and number of subselected queries $N_{\text{Q}}$, QUOKA has $\mathcal{O}((B_{\text{CP}} + (N_Q(1 + dn_{\text{KV}}))T)$ runtime complexity and $\mathcal{O}(n_{\text{KV}}N_QT)$ memory complexity. Note that both terms require $n_{\text{KV}}$ rather $n_{\text{Q}}$ where $n_{\text{Q}} > n_{\text{KV}}$, resulting in significant compute and memory savings. For SampleAttention, given the same parameters we obtain $\mathcal{O}((dn_{\text{Q}} + n_{\text{Q}}/n_{\text{KV}} + n_{\text{KV}})N_QT)$ runtime complexity and $\mathcal{O}(n_{\text{Q}}N_QT)$ memory complexity. Note that since this method

computes attention logits before aggregation, $n_Q$ appears in both terms. For Loki and SparQ, let $d_l < d$ be the lower dimension used to downproject the channels of $Q$ and $K$. With this SparQ has runtime complexity $\mathcal{O}(B_{CP}Td_ln_Q)$ and memory complexity $\mathcal{O}(n_QB_{CP}T)$ whereas Loki has runtime complexity $\mathcal{O}(d_ln_Q(B_{CP}T + d(B_{CP} + T)))$ due to the matrix multiplications and memory complexity $\mathcal{O}(n_QB_{CP}T)$. Note that Loki additionally has memory overhead of $\mathcal{O}(dd_ln_Q)$ to store the downprojection matrices in each layer. Finally, since LessIsMore is not applied uniformly across layers, where $L$ is the number of layers, we amortize complexity to obtain runtime complexity of $\mathcal{O}(dn_QB_{CP}T/L)$ and memory complexity of $\mathcal{O}(n_QB_{CP}T/L)$. Note that, asymptotically our method is more efficient than other methods due to the fact that $n_{KV} < n_Q$, justifying our pre-aggregation design.

Table 4: **Runtime and memory complexity of sparse attention methods.**

|  | Runtime Complexity | Memory Complexity |
|---|---|---|
| QUOKA | $\mathcal{O}(B_{CP} + (N_Q(1 + dn_{KV}))T)$ | $\mathcal{O}(n_{KV}N_QT)$ |
| SampleAttention | $\mathcal{O}((dn_Q + n_Q/n_{KV} + n_{KV})N_QT)$ | $\mathcal{O}(n_QN_QT)$ |
| SparQ | $\mathcal{O}(B_{CP}Td_ln_Q)$ | $\mathcal{O}(n_QB_{CP}T)$ |
| Loki | $\mathcal{O}(d_ln_Q(B_{CP}T + d(B_{CP} + T)))$ | $\mathcal{O}(n_QB_{CP}T)$ |
| LessIsMore | $\mathcal{O}(dn_QB_{CP}T/L)$ | $\mathcal{O}(n_QB_{CP}T/L)$ |

## D  MATHEMATICAL PROOFS

We provide the statement of Theorem 1 for completeness below along with the accompanying proof. Note that due to the unique geometry of LLM attention heads, this theorem implies that if $q_0$ is angularly distant from $M_Q$, it will have higher attention scores with most keys. Specifically, as $\text{CosSim}(M_Q, k) \to -1$, the cosine similarity $\text{CosSim}(M_Q, q^*)$ is bounded above by a monotonically decreasing function converging to $-\frac{1}{2}$.

**Theorem 1.** *Consider tokens a fixed query $q_0$ and key $k$, and let the average of a set of queries be denoted $M_Q$. Suppose $CosSim(k, q_0) = \beta_k > 0$ and $CosSim(M_Q, k) = \alpha_k < 0$. Then*

$$CosSim(M_Q, q^*) \leq 1 + \alpha_k\beta_k - 0.5\alpha_k^2 - 0.5\beta_k^2. \tag{A.1}$$

*Proof.* First compute the cosine similarity of $M_Q$ with $q^*$: $\text{CosSim}(M_Q, q^*) = \frac{q^{*\top}M_Q}{||q^*||||M_Q||}$. We can expand $M_Q$ in an orthonormal basis containing $k/||k||$, $\{k/||k||, r_1, ..., r_{n-1}\}$ where $M_Q = ||M_Q||\left(\alpha_qk/||k|| + \sum_{i=1}^{n-1}\alpha_ir_i\right)$ and $\alpha_i = \text{CosSim}(M_Q, r_i)$. Let $\beta_i = \text{CosSim}(q^*, r_i)$ and note since $r_i$ are orthonormal we have that $\alpha_k^2 + \sum_{i=1}^{n-1}\alpha_i^2 = 1$ and that $\beta_k^2 + \sum_{i=1}^{n-1}\beta_i^2 = 1$. Then

$$\frac{q^{*\top}M_Q}{||q^*||||M_Q||} = \frac{q^{*\top}\left(||M_Q||\alpha_kk/||k|| + \sum_{i=1}^{n-1}||M_Q||\alpha_ir_i\right)}{||q^*||||M_Q||}$$

$$= \alpha_k\beta_k + \frac{1}{||q^*||}\sum_{i=1}^{n-1}\alpha_iq^{*\top}r_i$$

$$= \alpha_k\beta_k + \sum_{i=1}^{n-1}\alpha_i\beta_i$$

$$\leq \alpha_k\beta_k + \sum_{i=1}^{n-1}|\alpha_i||\beta_i|$$

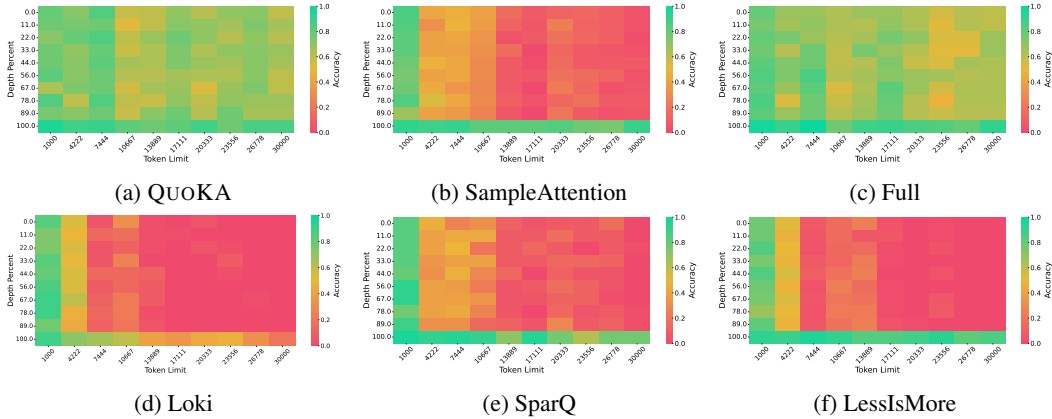

Figure 7: Accuracy across document length and needle depth for NIAH using additional sparse attention methods with $B_{\text{SA}} = 2048$ and $B_{\text{CP}} = 128$.

Using Young's inequality:

$$
\begin{aligned}
&\leq \alpha_k \beta_k + \frac{1}{2} \sum_{i=1}^{n-1} \alpha_i^2 + \beta_i^2 \\
&= \alpha_k \beta_k + \frac{1}{2}(1 - \alpha_k^2) + \frac{1}{2}(1 - \beta_k^2) \\
&= 1 + \alpha_k \beta_k - \frac{1}{2}\alpha_k^2 - \frac{1}{2}\beta_k^2
\end{aligned}
$$

$\square$

# E    ADDITIONAL NEEDLE IN A HAYSTACK EXPERIMENTS

In Figure 7 we show the results of the Needle In a Haystack experiment utilizing a variety of selective attention methods. We used the Llama3.2-3B-Instruct model and chunked prefill with $B_{\text{CP}} = 128$ with selective budget $B_{\text{SA}} = 2048$. The figure clearly shows the degradation incurred using most selective attention techniques with chunked prefill, while QUOKA is competitive even with full attention. This indicates that information is retained and recalled significantly more effectively using QUOKA under chunked prefill.

# F    ADDITIONAL RULER RESULTS

We provide additional results of QUOKA on the RULER benchmark, varying prompt length and selective attention budget $B_{\text{SA}}$. As shown in Table 5, QUOKA achieves near-dense accuracy even with $1/8 - 1/4$ of the full KV cache. This suggests that a significant speed up can be obtained for long context tasks with relative little performance drop.

# G    EXTENDED LONGBENCH RESULTS

In Tables 6 and 7 we present the performance on individual tasks of different models using different sparse attention methods across different selective attention budgets $B_{\text{SA}}$. In particular, Table 6 shows detailed task scores for different sparse attention methods across an array of models in which QUOKA consistently outperforms by a gap of $10 - 40\%$. Table 7 show a more detailed ablation studying the effects of varying $B_{\text{SA}}$ on LongBench tasks using only QUOKA. We can see that in general, a selective budget of 1024 achieves less than $5\%$ error overall, which is remarkable considering this is less than $1/10$ of the average task length for the considered LongBench tasks.

Table 5: **QUOKA RULER results** across prompt lengths and $B_{SA} \in \{1024, 2048, 4096\}$.

| Model | Budget | Prompt Length | | | |
| --- | --- | --- | --- | --- | --- |
| | | 4096 | 8192 | 16384 | 32768 |
| Llama-3.2-3B | Full | 87.50 | 81.33 | 76.98 | 74.31 |
| | 4096 | 87.51 | 80.60 | 75.69 | 69.95 |
| | 2048 | 87.33 | 80.20 | 73.79 | 63.67 |
| | 1024 | 86.71 | 80.19 | 70.90 | 57.01 |
| Llama-3.1-3B | Full | 89.45 | 90.24 | 90.53 | 86.65 |
| | 4096 | 89.45 | 89.03 | 87.46 | 81.33 |
| | 2048 | 89.20 | 88.82 | 86.21 | 78.86 |
| | 1024 | 89.24 | 89.51 | 83.25 | 70.04 |
| Qwen-2.5-3B | Full | 89.57 | 81.99 | 76.09 | 71.69 |
| | 4096 | 89.53 | 81.53 | 74.83 | 68.43 |
| | 2048 | 89.56 | 79.41 | 71.13 | 63.82 |
| | 1024 | 87.85 | 74.27 | 66.82 | 59.37 |
| Qwen-3-4B | Full | 93.32 | 91.68 | 91.18 | 88.54 |
| | 4096 | 93.32 | 91.83 | 91.06 | 87.68 |
| | 2048 | 93.13 | 91.80 | 88.87 | 85.23 |
| | 1024 | 93.73 | 91.07 | 88.57 | 74.83 |
| Smollm3 | Full | 91.21 | 83.46 | 80.11 | 75.18 |
| | 4096 | 91.21 | 83.70 | 79.66 | 71.83 |
| | 2048 | 90.96 | 83.06 | 78.26 | 67.65 |
| | 1024 | 89.97 | 79.94 | 72.69 | 61.37 |

## H MATH 500 RESULTS

The MATH-500 benchmark is a test designed to challenge the mathematical reasoning capabilities of LLMs. It comprises 500 problems sourced from high-level math competitions like the AMC and AIME, spanning domains such as algebra, geometry, number theory, precalculus, combinatorics, and probability. Unlike simpler datasets, MATH-500 emphasizes multi-step problem solving and abstract reasoning, requiring models to produce precise, step-by-step solutions. It has become a key benchmark for comparing LLMs' mathematical proficiency and due to long reasoning traces, is a good way to test the effectiveness of selective attention models during generation. While the majority of focus on QUOKA has been on its effectiveness with chunked prefill, it also performs remarkably well during decode as well. This is seen in Table 8 in which QUOKA performs as well or better than other competing methods, with much smaller reasoning traces generally resulting in significantly shorter reasoning times. This suggests that more information is collated during reasoning with QUOKA than competing methods.

## I ABLATIONS

Table 9: Scoring Ablation

| Scoring | Ruler Test Length | | | |
| --- | --- | --- | --- | --- |
| | 4096 | 8192 | 16384 | 32768 |
| Dot Prod. | 83.76 | 68.22 | 57.97 | 46.24 |
| Cos. Sim | **88.56** | **73.78** | **65.57** | **52.48** |

Table 10: Aggregation Ablation

| Aggr. | Ruler Test Length | | | |
| --- | --- | --- | --- | --- |
| | 4096 | 8192 | 16384 | 32768 |
| Mean | 84.28 | 69.14 | 58.81 | 45.53 |
| Max | **88.56** | **73.78** | **65.57** | **52.48** |

In this section we report the results of several ablation studies. In Table 9 we explore the effects of using cosine similarity instead of dot products to estimate the attention scores prior to aggregation in QUOKA. Specifically, we apply our method varying both scoring methods in the Llama3.2-3B-Instruct model on the RULER benchmark, where it is clear that the cosine similarity provides more normalized, better aggreated scores for selective attention. In Table 10 we demonstrate the differences between aggregation across the query dimension of the approximated attention scores using both the max and the mean. Again, the Llama3.2-3B-Instruct model and RULER benchmark were used, and

Table 6: **LongBench results** comparing performance of different selective attention mechanisms across different budgets.

| | | Single Doc. QA | | | Multi Doc. QA | | | Summarization | | | Fewshot Learning | | | Synthetic | Code | | |
| --- | --- | --- | --- | --- | --- | --- | --- | --- | --- | --- | --- | --- | --- | --- | --- | --- | --- |
| | | Narrative QA | Qasper | MF-en | HotpotQA | 2WikiMQA | Musique | GovReport | QMSum | MultiNews | TREC | TriviaQA | SAMSum | PR-en | Lcc | RB-P | Avg. |
| llama3.2-3B-Instruct | | 22.91 | 40.49 | 49.99 | 50.96 | 43.29 | 26.97 | 33.42 | 24.28 | 24.98 | 73.5 | 90.17 | 42.04 | 96.0 | 56.53 | 56.95 | 48.832 |
| QuoKA | 512 | 0.844 | 0.98 | 0.979 | 0.978 | 0.964 | 0.765 | 0.952 | 0.864 | 1.042 | 0.973 | 1.0 | 0.995 | 0.766 | 1.074 | 1.003 | 0.945 |
| | 1024 | 0.867 | 1.001 | 0.995 | 1.005 | 1.005 | 0.786 | 0.978 | 0.908 | 1.034 | 1.007 | 1.005 | 1.003 | 0.927 | 1.034 | 1.021 | 0.972 |
| | 2048 | 0.973 | 0.991 | 1.029 | 1.019 | 0.982 | 0.863 | 0.984 | 0.96 | 1.004 | 1.007 | 1.006 | 0.987 | 0.969 | 1.016 | 0.996 | 0.986 |
| SampleAttention | 512 | 0.759 | 0.66 | 0.647 | 0.751 | 0.645 | 0.276 | 0.917 | 0.787 | 1.028 | 0.667 | 0.983 | 0.977 | 0.12 | 0.962 | 0.896 | 0.738 |
| | 1024 | 0.814 | 0.817 | 0.79 | 0.764 | 0.694 | 0.502 | 0.953 | 0.832 | 1.019 | 0.741 | 0.982 | 0.971 | 0.234 | 0.972 | 0.916 | 0.8 |
| | 2048 | 0.919 | 0.922 | 0.958 | 0.94 | 0.801 | 0.753 | 0.963 | 0.915 | 1.008 | 0.864 | 1.0 | 1.006 | 0.531 | 0.995 | 0.942 | 0.901 |
| TidalDecode | 512 | 0.697 | 0.696 | 0.463 | 0.618 | 0.557 | 0.352 | 0.724 | 0.808 | 0.975 | 0.83 | 0.906 | 0.936 | 0.104 | 0.998 | 0.887 | 0.703 |
| | 1024 | 0.776 | 0.822 | 0.633 | 0.703 | 0.704 | 0.595 | 0.782 | 0.864 | 1.004 | 0.844 | 0.97 | 0.962 | 0.188 | 1.016 | 0.956 | 0.788 |
| | 2048 | 0.674 | 0.921 | 0.768 | 0.857 | 0.87 | 0.649 | 0.858 | 0.87 | 1.003 | 0.946 | 0.989 | 0.99 | 0.344 | 1.013 | 1.001 | 0.85 |
| SparQ | 512 | 0.679 | 0.607 | 0.635 | 0.659 | 0.604 | 0.346 | 0.909 | 0.787 | 1.045 | 0.68 | 0.948 | 0.973 | 0.089 | 0.983 | 0.88 | 0.721 |
| | 1024 | 0.75 | 0.801 | 0.779 | 0.734 | 0.795 | 0.566 | 0.936 | 0.817 | 1.027 | 0.755 | 0.985 | 0.976 | 0.224 | 0.977 | 0.907 | 0.802 |
| | 2048 | 0.909 | 0.927 | 0.964 | 0.932 | 0.773 | 0.748 | 0.967 | 0.898 | 1.002 | 0.844 | 0.986 | 0.996 | 0.536 | 0.989 | 0.923 | 0.893 |
| Loki | 512 | 0.57 | 0.662 | 0.634 | 0.662 | 0.575 | 0.211 | 0.891 | 0.803 | 1.038 | 0.646 | 0.908 | 0.933 | 0.052 | 0.927 | 0.783 | 0.686 |
| | 1024 | 0.878 | 0.799 | 0.683 | 0.687 | 0.649 | 0.371 | 0.923 | 0.831 | 1.022 | 0.707 | 0.947 | 0.952 | 0.104 | 0.967 | 0.835 | 0.757 |
| | 2048 | 0.907 | 0.897 | 0.85 | 0.837 | 0.837 | 0.554 | 0.947 | 0.884 | 1.004 | 0.844 | 0.973 | 0.998 | 0.234 | 0.985 | 0.873 | 0.842 |
| qwen2.5-3B-Instruct | | 21.4 | 35.52 | 37.33 | 29.3 | 24.44 | 17.64 | 32.56 | 21.9 | 23.29 | 67.5 | 85.37 | 43.85 | 47.5 | 51.57 | 47.75 | 39.128 |
| QuoKA | 512 | 0.68 | 0.833 | 0.937 | 0.834 | 0.811 | 0.578 | 0.948 | 0.903 | 1.002 | 0.889 | 0.915 | 0.977 | 0.751 | 0.974 | 1.006 | 0.869 |
| | 1024 | 0.713 | 0.992 | 1.022 | 1.018 | 0.841 | 0.924 | 0.984 | 0.942 | 0.99 | 0.985 | 0.983 | 0.97 | 0.858 | 0.952 | 0.995 | 0.945 |
| | 2048 | 0.974 | 0.962 | 1.02 | 0.998 | 0.813 | 1.023 | 0.998 | 0.973 | 1.0 | 1.022 | 0.972 | 0.992 | 0.879 | 1.009 | 1.018 | 0.977 |
| SampleAttention | 512 | 0.392 | 0.285 | 0.741 | 0.31 | 0.478 | 0.345 | 0.932 | 0.949 | 1.092 | 0.563 | 0.828 | 0.911 | 0.135 | 0.868 | 1.068 | 0.66 |
| | 1024 | 0.476 | 0.43 | 0.907 | 0.52 | 0.529 | 0.429 | 0.943 | 0.976 | 1.077 | 0.77 | 0.917 | 0.956 | 0.322 | 0.922 | 1.016 | 0.746 |
| | 2048 | 0.533 | 0.523 | 0.985 | 0.619 | 0.66 | 0.687 | 0.942 | 1.014 | 1.076 | 0.867 | 0.969 | 0.971 | 0.565 | 0.966 | 1.039 | 0.828 |
| TidalDecode | 512 | 0.172 | 0.188 | 0.422 | 0.179 | 0.382 | 0.159 | 0.569 | 0.726 | 0.967 | 0.526 | 0.307 | 0.642 | 0.057 | 0.846 | 0.766 | 0.461 |
| | 1024 | 0.245 | 0.273 | 0.654 | 0.249 | 0.426 | 0.162 | 0.67 | 0.794 | 1.035 | 0.681 | 0.456 | 0.775 | 0.107 | 0.917 | 0.901 | 0.556 |
| | 2048 | 0.34 | 0.418 | 0.691 | 0.368 | 0.564 | 0.234 | 0.814 | 0.864 | 1.068 | 0.904 | 0.693 | 0.882 | 0.094 | 0.981 | 0.967 | 0.659 |
| SparQ | 512 | 0.381 | 0.262 | 0.708 | 0.324 | 0.461 | 0.266 | 0.911 | 0.922 | 1.075 | 0.489 | 0.795 | 0.892 | 0.151 | 0.881 | 1.022 | 0.636 |
| | 1024 | 0.423 | 0.446 | 0.85 | 0.446 | 0.529 | 0.363 | 0.932 | 0.966 | 1.078 | 0.696 | 0.894 | 0.954 | 0.339 | 0.922 | 1.046 | 0.726 |
| | 2048 | 0.421 | 0.562 | 0.939 | 0.622 | 0.573 | 0.582 | 0.945 | 1.008 | 1.075 | 0.867 | 0.947 | 0.957 | 0.611 | 0.965 | 1.044 | 0.808 |
| Loki | 512 | 0.34 | 0.268 | 0.612 | 0.282 | 0.437 | 0.208 | 0.857 | 0.883 | 1.097 | 0.489 | 0.604 | 0.796 | 0.092 | 0.892 | 0.973 | 0.589 |
| | 1024 | 0.438 | 0.408 | 0.773 | 0.289 | 0.563 | 0.304 | 0.884 | 0.937 | 1.081 | 0.674 | 0.639 | 0.908 | 0.23 | 0.962 | 0.982 | 0.671 |
| | 2048 | 0.494 | 0.538 | 0.95 | 0.52 | 0.645 | 0.607 | 0.956 | 1.006 | 1.074 | 0.822 | 0.884 | 0.932 | 0.432 | 0.978 | 0.961 | 0.787 |
| smollm3 | | 19.46 | 37.83 | 43.11 | 18.42 | 19.58 | 10.02 | 34.18 | 23.1 | 26.78 | 76.0 | 84.86 | 44.94 | 80.96 | 67.19 | 63.1 | 43.302 |
| QuoKA | 512 | 0.89 | 0.919 | 1.003 | 1.243 | 1.201 | 0.915 | 0.942 | 0.924 | 0.991 | 0.921 | 1.031 | 0.974 | 1.017 | 1.008 | 1.006 | 0.998 |
| | 1024 | 0.955 | 0.985 | 1.007 | 1.313 | 1.162 | 1.109 | 0.981 | 0.977 | 0.981 | 0.947 | 1.02 | 0.981 | 1.032 | 1.01 | 0.996 | 1.03 |
| | 2048 | 0.917 | 1.024 | 1.012 | 1.244 | 1.135 | 1.135 | 0.988 | 0.99 | 0.996 | 0.974 | 1.014 | 1.006 | 0.989 | 1.006 | 0.996 | 1.028 |
| SampleAttention | 512 | 0.714 | 0.662 | 0.879 | 0.926 | 0.876 | 0.738 | 0.919 | 0.928 | 0.984 | 0.809 | 1.025 | 0.959 | 0.457 | 1.026 | 0.939 | 0.856 |
| | 1024 | 0.748 | 0.837 | 0.918 | 1.088 | 1.008 | 0.856 | 0.958 | 0.941 | 0.984 | 0.908 | 1.032 | 0.986 | 0.685 | 1.003 | 0.984 | 0.929 |
| | 2048 | 0.797 | 0.987 | 0.965 | 1.062 | 1.01 | 0.953 | 0.984 | 0.943 | 0.993 | 0.954 | 1.035 | 0.975 | 0.848 | 1.01 | 0.976 | 0.966 |
| TidalDecode | 512 | 0.684 | 0.615 | 0.715 | 0.72 | 0.689 | 0.637 | 0.815 | 0.883 | 0.96 | 0.842 | 0.887 | 0.951 | 0.21 | 0.967 | 0.902 | 0.765 |
| | 1024 | 0.761 | 0.778 | 0.808 | 0.742 | 0.812 | 0.658 | 0.893 | 0.9 | 0.987 | 0.888 | 0.976 | 0.965 | 0.544 | 0.976 | 0.937 | 0.842 |
| | 2048 | 0.831 | 0.936 | 0.903 | 0.844 | 0.868 | 0.738 | 0.941 | 0.939 | 0.993 | 0.961 | 1.007 | 0.962 | 0.884 | 0.981 | 0.982 | 0.918 |
| SparQ | 512 | 0.391 | 0.58 | 0.836 | 0.656 | 0.771 | 0.535 | 0.898 | 0.881 | 1.006 | 0.704 | 0.886 | 0.875 | 0.099 | 0.973 | 0.785 | 0.725 |
| | 1024 | 0.598 | 0.865 | 0.886 | 0.699 | 0.886 | 0.52 | 0.953 | 0.92 | 0.987 | 0.829 | 0.951 | 0.906 | 0.26 | 0.978 | 0.841 | 0.805 |
| | 2048 | 0.634 | 0.985 | 0.967 | 1.003 | 0.975 | 0.737 | 0.997 | 0.942 | 0.993 | 0.941 | 0.995 | 0.966 | 0.78 | 1.001 | 0.921 | 0.922 |
| Loki | 512 | 0.063 | 0.195 | 0.423 | 0.143 | 0.423 | 0.111 | 0.246 | 0.437 | 0.903 | 0.467 | 0.318 | 0.488 | 0.022 | 0.887 | 0.626 | 0.384 |
| | 1024 | 0.542 | 0.819 | 0.86 | 0.791 | 0.913 | 0.637 | 0.787 | 0.913 | 0.991 | 0.796 | 0.97 | 0.93 | 0.156 | 0.984 | 0.933 | 0.801 |
| | 2048 | 0.206 | 0.63 | 0.73 | 0.305 | 0.871 | 0.231 | 0.599 | 0.787 | 0.99 | 0.803 | 0.556 | 0.79 | 0.043 | 0.985 | 0.804 | 0.622 |
| qwen3-4B | | 28.02 | 43.75 | 53.46 | 55.55 | 43.87 | 31.82 | 32.48 | 24.81 | 25.08 | 73.5 | 88.26 | 43.69 | 96.5 | 64.14 | 59.02 | 50.93 |
| QuoKA | 512 | 0.814 | 0.964 | 0.996 | 0.974 | 0.89 | 1.054 | 0.996 | 0.94 | 1.032 | 0.946 | 1.008 | 1.005 | 0.876 | 0.985 | 1.009 | 0.966 |
| | 1024 | 0.914 | 0.986 | 1.033 | 0.988 | 1.016 | 0.963 | 0.992 | 0.963 | 1.02 | 0.993 | 1.024 | 1.028 | 0.959 | 0.987 | 1.012 | 0.992 |
| | 2048 | 0.876 | 0.992 | 1.024 | 1.015 | 0.977 | 0.995 | 0.989 | 0.988 | 1.008 | 0.993 | 1.011 | 1.023 | 1.026 | 1.008 | 1.004 | 0.995 |
| SampleAttention | 512 | 0.525 | 0.483 | 0.709 | 0.725 | 0.697 | 0.536 | 0.966 | 0.822 | 1.025 | 0.871 | 0.963 | 0.957 | 0.197 | 0.949 | 0.894 | 0.755 |
| | 1024 | 0.665 | 0.731 | 0.834 | 0.869 | 0.803 | 0.627 | 0.991 | 0.876 | 1.02 | 0.966 | 0.971 | 1.003 | 0.772 | 1.006 | 0.984 | 0.875 |
| | 2048 | 0.801 | 0.947 | 0.972 | 0.892 | 0.9 | 0.72 | 0.994 | 0.923 | 0.998 | 1.0 | 0.989 | 1.013 | 1.021 | 1.0 | 1.03 | 0.947 |
| TidalDecode | 512 | 0.401 | 0.513 | 0.597 | 0.583 | 0.582 | 0.395 | 0.774 | 0.842 | 0.96 | 0.796 | 0.76 | 0.915 | 0.047 | 0.936 | 0.878 | 0.665 |
| | 1024 | 0.483 | 0.682 | 0.731 | 0.709 | 0.728 | 0.474 | 0.866 | 0.879 | 0.996 | 0.871 | 0.917 | 0.966 | 0.358 | 0.996 | 0.931 | 0.773 |
| | 2048 | 0.537 | 0.83 | 0.831 | 0.903 | 0.781 | 0.586 | 0.925 | 0.919 | 1.01 | 0.952 | 0.968 | 0.982 | 0.803 | 0.996 | 1.002 | 0.868 |
| SparQ | 512 | 0.459 | 0.525 | 0.606 | 0.544 | 0.702 | 0.448 | 0.906 | 0.832 | 1.039 | 0.782 | 0.824 | 0.914 | 0.048 | 0.878 | 0.789 | 0.686 |
| | 1024 | 0.516 | 0.749 | 0.744 | 0.722 | 0.792 | 0.567 | 0.948 | 0.867 | 1.032 | 0.891 | 0.952 | 0.963 | 0.14 | 0.964 | 0.878 | 0.782 |
| | 2048 | 0.527 | 0.895 | 0.923 | 0.795 | 0.739 | 0.797 | 0.985 | 0.917 | 1.012 | 0.966 | 0.975 | 1.002 | 0.585 | 0.993 | 0.962 | 0.872 |
| Loki | 512 | 0.295 | 0.328 | 0.687 | 0.554 | 0.585 | 0.426 | 0.934 | 0.804 | 1.056 | 0.884 | 0.756 | 0.594 | 0.076 | 0.715 | 0.637 | 0.622 |
| | 1024 | 0.416 | 0.605 | 0.822 | 0.694 | 0.71 | 0.563 | 0.986 | 0.864 | 1.031 | 0.912 | 0.869 | 0.721 | 0.171 | 0.826 | 0.73 | 0.728 |
| | 2048 | 0.543 | 0.822 | 0.898 | 0.837 | 0.824 | 0.781 | 1.006 | 0.902 | 1.016 | 0.971 | 0.946 | 0.892 | 0.404 | 0.94 | 0.824 | 0.841 |

the higher scores indicated that the max captures important outlying key query interactions better than the mean.

In order to determine if QUOKA can be reliably used with different prefill chunk sizes, we use LongBench to test the Qwen3-4B model with both QUOKA and sample attention across different $B_{CP}$. The results are presented in Table 11 in which the number of queries selected $N_Q$ is 25% of $B_{CP}$ and the selctive budget $B_{SA} = 1024$. In this case we see no performance degradation when varying the prefill chunk size suggesting QUOKA is robust to changes in this parameter.

To explore the effects of varying the number of subselected queries during chunked prefill, we compared the performance of the Qwen3-4B model on the LongBench task using both QUOKA and SampleAttention. During prefill, the number of queries subselected varied between 4 and 128. These results are presented in Table 12 in which it is clear that very little performance is lost even with a small number of queries used to approximate attention scores for KV subselection. This performance, particularly compared to the closest competitor SampleAttention, seems to justify our query subselection method.

Table 7: **LongBench results (Higher is better)**. Each model utilizes chunked prefill with $B = 128$ with different selective budgets. Raw scores reported for base models and relative errors (percent of performance compared to baseline) presented for QUOKA.

| Method | Budget | Single Doc. QA | | | Multi Doc. QA | | | Summarization | | | Fewshot Learning | | | Synthetic | Code | | Avg. |
|---|---|---|---|---|---|---|---|---|---|---|---|---|---|---|---|---|---|
| | | Narrative QA | Qasper | MF-en | HotpotQA | 2WikiMQA | Musique | GovReport | QMSum | MultiNews | TREC | TriviaQA | SAMSum | PR-en | Lcc | RB-P | |
| llama3.1-8B-Instruct | | 31.16 | 46.94 | 56.45 | 58.03 | 48.24 | 31.97 | 34.79 | 25.55 | 26.94 | 73.0 | 91.72 | 43.2 | 99.5 | 62.06 | 52.55 | 52.14 |
| QUOKA | 256 | 0.713 | 0.909 | 0.89 | 0.823 | 0.97 | 0.874 | 0.958 | 0.895 | 1.001 | 0.836 | 0.997 | 1.027 | 0.879 | 1.079 | 1.021 | 0.925 |
| | 512 | 0.873 | 0.954 | 0.925 | 0.903 | 0.971 | 0.831 | 0.966 | 0.926 | 1.008 | 0.938 | 1.002 | 1.033 | 0.985 | 1.055 | 1.056 | 0.962 |
| | 1024 | 0.889 | 0.963 | 0.935 | 0.95 | 1.021 | 0.884 | 0.993 | 0.964 | 1.001 | 0.966 | 1.003 | 1.021 | 0.995 | 1.034 | 1.035 | 0.977 |
| | 2048 | 0.992 | 0.991 | 0.948 | 0.965 | 1.036 | 0.917 | 0.998 | 0.999 | 1.002 | 0.979 | 1.004 | 1.014 | 1.0 | 1.01 | 1.022 | 0.992 |
| | 4096 | 1.003 | 0.988 | 0.998 | 0.962 | 1.017 | 0.911 | 0.997 | 1.0 | 1.001 | 0.986 | 1.008 | 1.009 | 1.0 | 1.009 | 1.005 | 0.993 |
| llama3.2-3B-Instruct | | 22.91 | 40.49 | 49.99 | 50.96 | 43.29 | 26.97 | 33.42 | 24.28 | 24.98 | 73.5 | 90.17 | 42.04 | 96.0 | 56.53 | 56.95 | 48.832 |
| QUOKA | 256 | 0.773 | 0.959 | 0.907 | 0.86 | 0.825 | 0.677 | 0.931 | 0.854 | 1.034 | 0.837 | 0.982 | 0.979 | 0.396 | 1.031 | 0.952 | 0.866 |
| | 512 | 0.844 | 0.98 | 0.979 | 0.978 | 0.964 | 0.765 | 0.952 | 0.864 | 1.042 | 0.973 | 1.0 | 0.995 | 0.766 | 1.074 | 1.003 | 0.945 |
| | 1024 | 0.867 | 1.001 | 0.995 | 1.005 | 1.005 | 0.786 | 0.978 | 0.908 | 1.034 | 1.007 | 1.005 | 1.003 | 0.927 | 1.034 | 1.021 | 0.972 |
| | 2048 | 0.973 | 0.991 | 1.029 | 1.019 | 0.982 | 0.863 | 0.984 | 0.96 | 1.004 | 1.007 | 1.006 | 0.987 | 0.969 | 1.016 | 0.996 | 0.986 |
| | 4096 | 0.955 | 0.996 | 1.019 | 1.024 | 1.003 | 0.959 | 0.99 | 0.968 | 0.995 | 1.0 | 1.007 | 1.002 | 0.995 | 0.999 | 0.997 | 0.994 |
| qwen2.5-7B-Instruct | | 25.84 | 37.79 | 43.02 | 47.66 | 40.11 | 25.22 | 33.52 | 22.49 | 23.12 | 66.5 | 87.46 | 44.71 | 96.25 | 57.76 | 61.77 | 47.548 |
| QUOKA | 256 | 0.594 | 0.588 | 0.736 | 0.616 | 0.565 | 0.433 | 0.933 | 0.835 | 0.959 | 0.812 | 0.959 | 0.975 | 0.299 | 0.854 | 0.692 | 0.723 |
| | 512 | 0.811 | 0.904 | 0.98 | 0.85 | 0.798 | 0.721 | 0.958 | 0.884 | 0.98 | 0.932 | 0.998 | 1.0 | 0.777 | 0.966 | 0.865 | 0.895 |
| | 1024 | 0.803 | 0.983 | 0.97 | 0.996 | 0.84 | 0.856 | 0.97 | 0.954 | 0.996 | 0.925 | 0.977 | 1.02 | 0.921 | 0.99 | 0.94 | 0.943 |
| | 2048 | 0.894 | 0.927 | 0.97 | 1.001 | 0.856 | 0.851 | 0.987 | 0.98 | 0.999 | 0.947 | 0.989 | 1.029 | 0.949 | 1.007 | 0.967 | 0.957 |
| | 4096 | 0.937 | 0.982 | 1.032 | 1.0 | 0.931 | 0.905 | 0.987 | 1.004 | 0.998 | 1.0 | 0.992 | 1.014 | 0.968 | 1.001 | 0.99 | 0.983 |
| qwen2.5-3B-Instruct | | 21.4 | 35.52 | 37.33 | 29.3 | 24.44 | 17.64 | 32.56 | 21.9 | 23.29 | 67.5 | 85.37 | 43.85 | 47.5 | 51.57 | 47.75 | 39.128 |
| QUOKA | 256 | 0.495 | 0.548 | 0.711 | 0.459 | 0.705 | 0.337 | 0.891 | 0.844 | 0.991 | 0.644 | 0.782 | 0.937 | 0.563 | 0.92 | 0.925 | 0.72 |
| | 512 | 0.68 | 0.833 | 0.937 | 0.834 | 0.811 | 0.578 | 0.948 | 0.903 | 1.002 | 0.889 | 0.915 | 0.977 | 0.751 | 0.974 | 1.006 | 0.869 |
| | 1024 | 0.713 | 0.992 | 1.022 | 1.018 | 0.841 | 0.924 | 0.984 | 0.942 | 0.99 | 0.985 | 0.983 | 0.97 | 0.858 | 0.952 | 0.995 | 0.945 |
| | 2048 | 0.974 | 0.962 | 1.02 | 0.998 | 0.813 | 1.023 | 0.998 | 0.973 | 1.0 | 1.022 | 0.972 | 0.992 | 0.879 | 1.009 | 1.018 | 0.977 |
| | 4096 | 0.943 | 1.04 | 1.034 | 0.968 | 0.913 | 1.022 | 0.998 | 0.997 | 1.001 | 0.985 | 0.994 | 0.994 | 0.911 | 0.985 | 0.99 | 0.985 |
| qwen3-8B | | 26.44 | 47.73 | 53.7 | 59.34 | 43.45 | 34.77 | 33.33 | 24.13 | 24.94 | 71.5 | 90.71 | 44.33 | 100.0 | 69.01 | 62.0 | 52.359 |
| QUOKA | 256 | 0.696 | 0.819 | 0.859 | 0.763 | 0.795 | 0.572 | 0.993 | 0.848 | 0.986 | 0.93 | 0.978 | 0.985 | 0.885 | 0.973 | 0.895 | 0.865 |
| | 512 | 0.89 | 0.921 | 0.987 | 1.007 | 0.938 | 0.868 | 1.005 | 0.951 | 0.998 | 1.007 | 0.996 | 1.01 | 1.0 | 1.009 | 1.03 | 0.974 |
| | 1024 | 0.928 | 0.999 | 0.998 | 0.987 | 0.954 | 0.903 | 1.002 | 0.981 | 1.001 | 1.007 | 1.006 | 1.023 | 1.0 | 1.007 | 1.034 | 0.988 |
| | 2048 | 0.958 | 0.992 | 1.014 | 1.024 | 0.932 | 0.897 | 1.007 | 0.99 | 1.001 | 1.007 | 0.999 | 1.014 | 1.0 | 0.999 | 1.008 | 0.989 |
| | 4096 | 0.965 | 0.996 | 0.999 | 1.024 | 0.962 | 0.985 | 1.003 | 0.998 | 1.0 | 1.0 | 1.0 | 0.998 | 1.0 | 0.996 | 1.003 | 0.995 |
| qwen3-4B | | 28.02 | 43.75 | 53.46 | 55.55 | 43.87 | 31.82 | 32.48 | 24.81 | 25.08 | 73.5 | 88.26 | 43.69 | 96.5 | 64.14 | 59.02 | 50.93 |
| QUOKA | 256 | 0.627 | 0.884 | 0.918 | 0.789 | 0.804 | 0.717 | 0.99 | 0.88 | 1.02 | 0.789 | 0.98 | 0.947 | 0.741 | 0.937 | 0.925 | 0.863 |
| | 512 | 0.814 | 0.964 | 0.996 | 0.974 | 0.89 | 1.054 | 0.996 | 0.94 | 1.032 | 0.946 | 1.008 | 1.005 | 0.876 | 0.985 | 1.009 | 0.966 |
| | 1024 | 0.914 | 0.986 | 1.033 | 0.988 | 1.016 | 0.963 | 0.992 | 0.963 | 1.02 | 0.993 | 1.011 | 1.028 | 0.959 | 0.987 | 1.012 | 0.992 |
| | 2048 | 0.876 | 0.992 | 1.024 | 1.015 | 0.977 | 0.995 | 0.989 | 0.988 | 1.008 | 0.993 | 1.011 | 1.023 | 1.026 | 1.008 | 1.004 | 0.995 |
| | 4096 | 0.91 | 1.001 | 1.011 | 1.009 | 0.997 | 0.989 | 1.0 | 1.002 | 1.007 | 1.007 | 1.0 | 0.991 | 1.0 | 1.0 | 1.003 | 0.996 |
| qwen3-1.7B | | 18.94 | 25.17 | 46.48 | 39.07 | 32.52 | 18.06 | 30.73 | 22.88 | 24.77 | 73.5 | 85.39 | 42.37 | 94.0 | 44.6 | 37.81 | 42.419 |
| QUOKA | 256 | 0.801 | 0.947 | 0.843 | 0.711 | 0.8 | 0.587 | 0.988 | 0.91 | 1.007 | 0.864 | 0.933 | 0.959 | 0.646 | 0.832 | 0.726 | 0.837 |
| | 512 | 0.987 | 0.986 | 0.948 | 0.88 | 0.984 | 0.822 | 1.032 | 0.97 | 1.027 | 0.946 | 1.01 | 0.978 | 0.949 | 0.933 | 0.841 | 0.953 |
| | 1024 | 1.041 | 1.003 | 0.983 | 0.973 | 1.052 | 0.865 | 1.021 | 0.996 | 1.01 | 0.98 | 1.017 | 0.999 | 0.979 | 1.017 | 0.834 | 0.984 |
| | 2048 | 0.968 | 0.995 | 0.967 | 0.976 | 1.015 | 1.007 | 1.005 | 1.003 | 1.0 | 0.98 | 1.013 | 1.0 | 1.0 | 1.001 | 0.869 | 0.986 |
| | 4096 | 0.982 | 1.019 | 0.987 | 0.957 | 1.026 | 1.048 | 0.988 | 1.025 | 0.997 | 1.0 | 1.006 | 1.014 | 0.995 | 1.0 | 0.973 | 1.001 |
| qwen3-30B-A3B-2507 | | 31.33 | 42.19 | 55.0 | 63.0 | 55.94 | 33.85 | 31.06 | 21.79 | 23.58 | 76.72 | 90.42 | 46.41 | 100.0 | 74.33 | 67.08 | 54.18 |
| QUOKA | 256 | 0.658 | 1.048 | 0.887 | 0.862 | 0.604 | 0.61 | 1.011 | 0.904 | 0.978 | 0.899 | 0.987 | 1.0 | 1.0 | 0.979 | 0.859 | 0.886 |
| | 512 | 0.903 | 1.022 | 0.95 | 0.988 | 0.897 | 0.964 | 1.006 | 0.976 | 0.993 | 0.971 | 1.002 | 1.003 | 1.0 | 1.007 | 0.992 | 0.978 |
| | 1024 | 0.991 | 1.015 | 0.953 | 1.005 | 0.956 | 0.985 | 1.0 | 1.007 | 1.003 | 0.997 | 1.015 | 1.009 | 1.0 | 1.002 | 1.005 | 0.996 |
| | 2048 | 0.992 | 1.021 | 0.98 | 1.005 | 0.974 | 1.012 | 1.009 | 1.002 | 0.998 | 0.997 | 1.015 | 1.012 | 1.0 | 0.996 | 1.009 | 1.001 |
| | 4096 | 1.011 | 1.007 | 0.983 | 1.003 | 1.005 | 0.974 | 1.002 | 0.997 | 0.996 | 0.997 | 1.007 | 0.994 | 1.0 | 0.999 | 1.004 | 0.999 |
| smollm3 | | 19.46 | 37.83 | 43.11 | 18.42 | 19.58 | 10.02 | 34.18 | 23.1 | 26.78 | 76.0 | 84.86 | 44.94 | 80.96 | 67.19 | 63.1 | 43.302 |
| QUOKA | 256 | 0.777 | 0.847 | 0.919 | 1.127 | 1.097 | 0.848 | 0.937 | 0.928 | 0.979 | 0.757 | 1.009 | 0.939 | 0.704 | 0.965 | 0.894 | 0.915 |
| | 512 | 0.89 | 0.919 | 1.003 | 1.243 | 1.201 | 0.915 | 0.942 | 0.924 | 0.991 | 0.921 | 1.031 | 0.974 | 1.017 | 1.008 | 0.987 | 0.998 |
| | 1024 | 0.955 | 0.985 | 1.007 | 1.313 | 1.162 | 1.109 | 0.981 | 0.977 | 0.981 | 0.947 | 1.02 | 0.981 | 1.032 | 1.01 | 0.996 | 1.03 |
| | 2048 | 0.917 | 1.024 | 1.012 | 1.244 | 1.135 | 1.135 | 0.988 | 0.99 | 0.996 | 0.974 | 1.014 | 1.006 | 0.989 | 1.006 | 0.996 | 1.028 |
| | 4096 | 0.891 | 1.002 | 1.008 | 1.226 | 1.046 | 1.004 | 1.016 | 0.997 | 0.995 | 1.0 | 0.993 | 0.999 | 1.023 | 0.999 | 0.996 | 1.013 |

Table 8: **Math 500 results** across different selective budgets utilizing QUOKA. Reasoning generation length limited to 8192 tokens.

| | Budget | Flex Match | Exact Match | Avg. Gen. Length |
|---|---|---|---|---|
| GPT-OSS-20b | | 0.893 | 0.694 | 2558.262 |
| SparQ | 128 | 0.744 | 0.569 | 3023.71 |
| | 256 | 0.848 | 0.648 | 2436.86 |
| Loki | 128 | 0.905 | **0.711** | 2276.0 |
| | 256 | 0.903 | **0.714** | 2181.49 |
| LessIsMore | 128 | 0.782 | 0.62 | 3591.4 |
| | 256 | 0.85 | 0.681 | 2830.49 |
| QUOKA | 128 | **0.911** | 0.707 | 2199.93 |
| | 256 | **0.913** | 0.711 | 2539.11 |
| Qwen3-4B | | 0.82 | 0.701 | 1179.78 |
| SparQ | 128 | 0.727 | 0.615 | 2044.902 |
| | 256 | 0.777 | 0.658 | 1518.82 |
| Loki | 128 | 0.785 | 0.668 | 1345.88 |
| | 256 | 0.821 | 0.702 | 1143.51 |
| LessIsMore | 128 | 0.729 | 0.621 | 2674.25 |
| | 256 | 0.814 | 0.695 | 1618.00 |
| QUOKA | 128 | **0.828** | **0.708** | 1119.46 |
| | 256 | **0.823** | **0.703** | 1156.12 |
| Smollm3 | | 0.659 | 0.522 | 629.04 |
| SparQ | 128 | 0.647 | 0.514 | 715.18 |
| | 256 | **0.665** | **0.532** | 634.34 |
| Loki | 128 | 0.615 | 0.494 | 621.02 |
| | 256 | 0.654 | 0.52 | 618.47 |
| LessIsMore | 128 | 0.623 | 0.495 | 642.26 |
| | 256 | 0.657 | 0.527 | 598.907 |
| QUOKA | 128 | **0.66** | **0.521** | 578.10 |
| | 256 | 0.661 | 0.527 | 614.73 |

Table 11: **LongBench results for the Qwen3-4B model utilizing QUOKA across** $B_{\text{CP}} \in \{128, 256, 512\}$ **with** $B_{\text{SA}} = 1024$ **(Higher is better).** Set $N_q$ to 25% of $B_{\text{CP}}$. Relative scores compared to the dense model are reported.

| $B_{\text{CP}}$ | 128 | 256 | 512 |
|---|---|---|---|
| QUOKA | **0.978** | **0.982** | **0.977** |
| Sample Attention | 0.876 | 0.871 | 0.872 |

Table 12: **LongBench results for the Qwen3-4B model utilizing QUOKA across** $N_Q \in \{4, 8, 16, 32, 64, 128\}$ **with** $B_{\text{SA}} = 1024$ **and** $B_{\text{CP}} = 128$ **(Higher is better).** Relative scores compared to the dense model are reported.

| $N_Q$ | 4 | 8 | 16 | 32 | 64 | 128 |
|---|---|---|---|---|---|---|
| QUOKA | **0.935** | **0.957** | **0.965** | **0.978** | **0.988** | **0.995** |
| Sample Attention | 0.854 | 0.862 | 0.88 | 0.878 | 0.885 | 0.884 |

