# OpenReview forum: "QuoKA: Query-Oriented KV Selection for Efficient LLM Prefill"
_ICLR.cc/2026/Conference — ICLR 2026 Poster_

### Official Review · Reviewer_JhZf · 2025-10-27

**Soundness:** 2
**Presentation:** 3
**Contribution:** 3
**Rating:** 6
**Confidence:** 2

**Summary:**

The authors present QUOKA, a training-free sparse attention algorithm for efficient LLM prefill. The method prioritizes queries with low cosine similarity to the mean to subselect keys. This achieves a 3-7x speedup and an 88% reduction in key-value pairs, while maintaining near-baseline accuracy.

**Strengths:**

-  The paper introduces a heuristic for query selection based on the hypothesis that queries with greater angular distance from the mean are more informative. This provides a geometric perspective on the multi-query attention problem that is distinct from conventional approaches focused on representativeness.
- The algorithm is designed to be training-free and hardware-agnostic by avoiding the use of custom kernels. This design allows for its potential integration into various inference systems without requiring model retraining or fine-tuning.
- The paper's experiments report reductions in Time-to-First-Token and attention latency across multiple hardware platforms. The results also show that these efficiency improvements are achieved while task accuracy is maintained close to that of the dense attention baseline.

**Weaknesses:**

- The paper does not include a comparison to alternative query selection strategies, such as selecting representative queries via clustering (e.g., K-Means centroids). Without this comparison, it is difficult to fully assess the performance of the proposed "outlier query" heuristic relative to more established methods for summarization.
- The evaluation of the method's applicability to generation tasks appears less developed. Since the core query subselection component is bypassed in the single-query decoding scenario, a comparison against baselines specifically designed for decoding-phase KV management would be necessary to fully substantiate the method's competitiveness in this setting.
- The potential impact of quantization on the method's performance is not discussed. The algorithm's reliance on precise geometric relationships (via cosine similarity) means its robustness in low-bit precision environments, which are common on its target hardware, remains an important but unevaluated factor.

**Questions:**

- Could you provide a more direct comparison or discussion against a "central representativeness" approach, such as selecting K-Means query centroids? This would help to empirically situate the performance of your proposed heuristic.
- The assumptions in Theorem 1 are central to the method's motivation. How consistently do these geometric conditions hold empirically across different models and layers? Supporting statistics or visualizations would be valuable.
- For the Math500 experiments where query subselection is not applied, could you please clarify the exact mechanism of QUOKA? Specifically, how are keys selected for the single active query during the generation phase?

---

> ### Author Response · Authors · 2025-11-17
>
> We would like to thank you for taking the time to review our paper. We will try to address your concerns in order:
>
> 1.)	Your suggestion that we compare our query subselection against using k-means to utilize “representative” queries is very interesting. To test this idea, we replaced steps 1-5 in algorithm 1 with kmeans. We tested this method on longbench for qwen3-4B and llama3.2-3B using k means with 16 centroids and a tolerance (stopping condition) epsilon = .01. We present the results below using a selective budget of 1024. We find that these results lag slightly behind the performance of the original method (although they are very close).
>
> More importantly, k means has a high computational overhead, and in timing tests this method is significantly 1.5-2x slower than the original algorithm. While I think there is lots of potential to use k-means to compute representative tokens for attention (see https://arxiv.org/pdf/2505.02922) in this particular case, since chunked prefill requires repeated application of kmeans it may not be the most efficient choice. However, I do think if it was possible to precalibrate fixed centroids for each layer/head, this technique could yield very good performance. Indeed, we believe that this could allow for fewer “representative” queries to be used in the importance score computation, and hence a much more efficient algorithm if k-means (or another similar method) can be implemented efficiently.
>
> | Model + Method | narrativeqa | qasper | multifieldqa_en | multifieldqa_zh | hotpotqa | 2wikimqa | musique | dureader | gov_report | qmsum | multi_news | vcsum | trec | triviaqa | samsum | lsht | passage_count | passage_retrieval_en | passage_retrieval_zh | lcc | repobench-p | Average Score |
> |---------------|------------|--------|-----------------|-----------------|----------|----------|---------|----------|------------|-------|------------|-------|------|----------|--------|------|---------------|-----------------------|-----------------------|-----|-------------|---------------|
> | Llama3.2-3B Quoka (Original)  | 21.72 | 39.13 | 47.6 | 51.74 | 46.7 | 37.6 | 17.74 | 30.99 | 32.81 | 22.33 | 25.91 | 15.71 | 64.0 | 90.58 | 41.31 | 28.75 | 7.14 | 73.5 | 13.0 | 53.46 | 53.47 | 38.81 |
> | Llama3.2-3B Quoka (KMeans)  | 21.64 | 40.27 | 46.61 | 52.46 | 46.05 | 36.86 | 16.44 | 31.32 | 32.88 | 22.28 | 25.98 | 15.87 | 67.5 | 90.17 | 41.76 | 29.25 | 5.5 | 53.0 | 13.0 | 53.13 | 53.25 | 37.87 |
> | Qwen3-4B Quoka (Original) | 24.14 | 42.85 | 54.34 | 66.1 | 56.18 | 41.01 | 33.52 | 29.48 | 32.28 | 23.62 | 25.31 | 15.12 | 71.5 | 89.35 | 44.29 | 40.0 | 1.75 | 95.0 | 98.5 | 62.96 | 58.22 | 47.88 |
> | Qwen3-4B Quoka (Kmeans) | 25.46 | 40.63 | 53.68 | 64.0 | 49.96 | 40.45 | 28.66 | 29.05 | 32.04 | 23.68 | 25.39 | 15.35 | 74.0 | 88.04 | 44.05 | 40.0 | 0.75 | 94.5 | 98.0 | 62.92 | 60.37 | 47.19 |
>
>
> 2.)	This is an important consideration. We had not computed the percentage of time that the condition in theorem 1 is satisfied. In order to test this, we passed a single sample from the narrativeqa dataset into several models (llama3.2-3B-instruct, qwen3-4B and smollm3) and computed the percentage of heads in which there existed a query which satisfied the specifications of the theorem. For Llama3.2, this represented 78% of the heads across the model, 83% for qwen3-4B, and 65% for smollm3. It is worth noting that this token often corresponded to the “sink token” but that the general intuition that queries with generally higher attention scores that are not concentrated on a small number of keys have higher cosine similarity with most keys due to the relationship between logit computation and cosine similarity.  Moreover, even in the heads/layers where this assumption is not satisfied, the heuristic appears to work well.
>
> 3.)	This is a very good question and I’m sorry this wasn’t more clear from the paper. It essentially boils down to step 6-8 in algorithm one (which may be the most important result of the paper.) We aren’t subselecting queries since there is only one query in the case of generation (or reasoning as in math500). However, we can efficiently compute importance scores by aggregating queries across KV heads prior to importance score computation. This reduces the required computation by a factor of (number of heads Q / number of heads K). Moreover, since values are not used in the importance score computation, we reduce our memory bandwidth requirements for values by a significant amount since we can move a small subset of these onto the relevant device for the attention operation calculation. Since most models use GQA, the number of Q heads is significantly less than the number of KV heads. To use Qwen3-4B as a specific example, we would require almost 4x less computation to compute the importance score, then subselect the KV pairs for use in attention later.

---

### Official Review · Reviewer_nDFH · 2025-10-31

**Soundness:** 3
**Presentation:** 3
**Contribution:** 2
**Rating:** 4
**Confidence:** 4

**Summary:**

Due to the quadratic nature of attention, long-context prefill remains a major challenge for large language models (LLMs). However, each query only needs to attend to a subset of key–value (KV) pairs to achieve reasonable performance. Existing sparse attention methods either rely on fixed attention patterns or are optimized primarily for the generation phase.
This paper presents **QUOKA**, which identifies representative queries from each chunk based on the smallest cosine similarity to the mean query vector, then uses these representative queries to locate important KV pairs. The model finally performs full query–subset KV attention. QUOKA achieves strong performance on long-context benchmarks and is efficient to implement.

**Strengths:**

- The evaluation is comprehensive and the reported results are impressive.
- The proposed method is clearly described, and the paper is easy to follow.

**Weaknesses:**

- Hardware efficiency may degrade due to the small chunk size and discontinuous KV selection.

**Questions:**

Thanks for submitting to ICLR 2026. This paper introduces an interesting idea of filtering query vectors using cosine similarity, inspired by DiffKV’s approach to KV cache filtering. However, I still have some concerns about the motivation and the efficiency claims.

## 1. Intuition behind the “critical” query vectors
The intuition for using “critical” query vectors is not fully convincing. It is true that such queries are closer to the key vector space and may attend to a wider range of keys or exhibit higher variance in attention scores. However, since the softmax operation is applied independently to each query, the proposed approach only ensures that these selected queries have smaller attention errors. It does not necessarily guarantee that other queries in the same group will also exhibit small errors. Intuitively, it is unclear why these tokens should be more important for overall generation accuracy.

## 2. Limited benefit in the generation phase
It is unclear how this method can lead to meaningful speedups during the generation phase. QUOKA estimates the similarity between each query and all keys, but since generation involves only one query vector, this step effectively performs half of the full attention computation. Even after selecting top-k keys and multiplying by the corresponding values, the overall computational reduction—and thus the speedup—appears minimal.

## 3. Hardware inefficiency due to small chunks
In the evaluation, the block size is set to 128. However, this configuration is inefficient on modern hardware, as each GEMM or attention operation on such small blocks yields low arithmetic intensity and thus lower TFLOPs. This effect is particularly noticeable on H100 and B200. In your latency test, do you also use block size 128 for the full attention baseline? A fairer comparison would allow full attention to use larger block sizes (e.g., 1024 or 2048), which are more hardware-efficient.

## 4. Constraints on KV selection
Are there any constraints imposed on the selected KV pairs? If the selected KVs are discontinuous, how is self-attention computed efficiently using existing kernels? Discontinuous memory access patterns can severely hinder performance unless carefully optimized.

---

> ### Author Response · Authors · 2025-11-17
>
> We would like to thank reviewer nDFH for their time and effort reading and reviewing our paper. We would like to address your concerns one by one:
>
> 1.)	The query subselection mechanism was designed to reduce the the amount of computation required to estimate importance scores for KV-cache subselection. We found that many queries tend to interact with keys similarly, and thus introduced redundancy in the score calculation. The method we arrived at to reduce the number of queries in that score calculation was designed to reduce that redundancy (by selecting the most geometrically disparate tokens) and make the score calculation for KV subselection more efficient.
>
> 2.)	This is a critical point that I apologize was not made more clear in the paper. In essence, since attention is often very sparse, and not all KV pairs are used in each attention computation, we are wasting computation/memory transfer on irrelevant keys/values. Thus we want to subselect those KV pairs that are relevant to each attention computation to increase attention computation efficiency. This selection mechanism would ideally be as efficient as possible as well. The structure of the selection mechanism is quite simple in our case, we compute an importance score for each KV-pair, which has shape (batch size, number of heads K, number of K tokens). We compute this score as the aggregated cosine similarity between keys and queries CosSim(K, Q). Naively (ignoring batch size wlog) the computation of this quantity requires H_Q * N_Q * N_K ops (where N_Q, and N_K are the number of keys and queries and H_Q is the number of query heads, see table 4 for more details). Steps 1-5 of algorithm 1 reduce this computation by reducing the N_Q. However your question pertains to generation where N_Q = 1. This is where step 8 of algorithm 1 makes our method much more efficient than simply computing attention. We utilize the commutativity of the dot product with sums and aggregate normalized queries among the heads prior to the score calculation CosSim(K, Q). The score calculation then only requires (H_K * N_K) operations for generation (H_K is the number of key heads), which is significantly less than the (H_Q *N_K) operations for the attention matrix computation (since most models use GQA and H_K < H_Q). Moreover, since the score calculation does not require values, we reduce the memory bandwidth significantly since only a fraction of the values are utilized by the final attention calculation after KV pair subselection.
>
> 3.)	Our goal was to design an algorithm that performs well with chunked prefill on edge devices. As such, we focused on configurations for chunked prefill that are efficient for such hardware, in this case using chunk size 128. While we believe that our method would be highly performative with higher chunk sizes (and we provide some ablation studies in the appendix to demonstrate this point) this was not the main focus of the paper.
>
> 4.)	There are no constraints on KV subselection, although such constraints such as selecting blocks of contiguous tokens could easily be incorporated. However, we found the algorithm to be highly efficient, as shown in our results, even without this consideration. With more optimized memory management we expect even greater gains in efficiency.
>
> Once again, we would like to thank the reviewer for their time and effort.

---

### Official Review · Reviewer_gdtt · 2025-10-31

**Soundness:** 2
**Presentation:** 3
**Contribution:** 2
**Rating:** 6
**Confidence:** 4

**Summary:**

The authors propose a better chunked prefill technique which only uses a small subset of KV cache for each chunked prefill. The algorithm goes like this -- choose a subset of queries to select KV cache, compute scores for these queries and aggregate across heads and queries, choose topk scoring KV for chunked prefill. They show that their method outperforms a bunch of baselines at same sparsity.

**Strengths:**

Strengths
1. Great experimental breadth and strong performance compared to baselines. The experiments cover multiple benchmarks and baselines.
2. Easy to understand algorithm.

**Weaknesses:**

1. The approach is not very principled. While it is true that queries with high similarity with K will have low similarity with Mean(Q) due to OOD nature of query and key distributions (this is what theorem says), the converse is not true (this is what you want for efficiency) . It is especially not true in high dimensions -- where it is highly likely that queries with low similarity with Mean(Q) would also have low similarity with K.

So this being a critical component of algorithm is unsettling. I would assume that most queriers chosen are actually even worse than Mean(Q) w.r.t similarity with K.  Can we have distribution plots of cosine similarities of chosen queries vs. all the queries.

Having said that their experimental section strongly supports their method.

2. some latest baselines are missing -- duoattention, xattention, spargeattention, might be good to add discussion / results for these.

**Questions:**

1. Can we have plots for cosine similarities (K, q) for chosen queries and all the queries.

---

> ### Author Response · Authors · 2025-11-17
>
> The authors would like to thank reviewer gdtt for their insightful and thoughtful review. The concern about the principled nature of the query selection mechanism central to the efficacy of the algorithm is certainly a valid point to bring up. It is certainly true that in high dimensions, queries having low cosine similarity with mean(Q) does not guarantee a higher cosine similarity with keys. However, empirical evidence suggests that queries that fall outside the "typical" query distribution tend to be more aligned to the key distribution (additionally the key and query distributions tend to be highly clustered in terms of cosine similarity.) We suspect that this allows the models to selectively emphasize a small number of tokens through attention scores. The paper (https://arxiv.org/pdf/2504.15364) offers a more comprehensive empirical examination of this phenomena.
>
> With all that being said, we would be more than happy to provide plots for cosine similarities (K,q) in the final paper. Additionally to your point 2.) We chose not to compare our method against xattention, since we didn't believe it was applicable to our primary use case utilizing chunked prefill, and spargeattention requires different custom kernels across different hardware. However, DuoAttention is certainly something we would like to contrast with our method and we will endeavor to implement and compare these methods in the future.
>
> Thank you very much again for the time and effort reading our paper. If you have further questions or concerns, please don't hesitate to ask.

---

### Official Review · Reviewer_gbPS · 2025-11-01

**Soundness:** 3
**Presentation:** 2
**Contribution:** 3
**Rating:** 8
**Confidence:** 4

**Summary:**

The paper proposes a novel sparse attention method for improving the efficiency of transformer models during the prefill stage with chunked prefill. The idea is to select a subset of important queries by divergence to the mean query and only compute attention for those important queries and important key-value pairs. The method is evaluated on SOTA open-source LLMs against other sparse attention baselines and shows better efficiency and performance frontier.

**Strengths:**

- The proposed method is well-motivated with insight experiments
- The pseudo-code is extremely helpful in understanding the method
- The experiment is solid

**Weaknesses:**

- Some explanations of the claims are confusing

**Questions:**

Thank you for your submission. I like the paper overall and think the method is well motivated. I particulary enjoy the insight experiments in Figure 2 and 3, which make the motivation very clear. However, some claims and descriptions in the paper are poorly explained and a bit vague to me. I would appreciate it if the authors can clarify the following questions:
- What is the relationship with chunked prefill? The proposed method seems to be highly dependent on chunked prefill, but the relationship is not very clear to me. I can understand that some previous sparse attention methods can be inefficient under prefill with multiple queries due to aggregated sparsity. But why is chunked prefill specifically needed for the proposed method? Is it possible to use the proposed method without chunked prefill?
- "however, due to dynamic compute graph and KV cache memory bandwidth overhead under chunked prefill, their benefits are limited." Can you please elaborate more on this point? Why dynamic compute graph and KV cache memory bandwidth overhead limit the benefits?
- "During prefill, when relevant KVs are selected for many queries at once, this can result in significant performance degradations. Under chunked prefill, where important KVs are repeatedly subselected for multiple queries, these degradations become more pronounced." These two sentences are particulary confusing to me. Why chunked prefill makes the performance degradation more pronounced?
- The gather operator in algorithm 1 has inconsistent notations (at line 4 and line 12).
- “As discussed in Section 2, existing sparse attention methods face limitations in prefill efficiency and portability.” What do you mean by portability here?
- I don't fully understand the query selection process. Many of the previous sparse attention methods also reduce among the KV dimension, so the attention socres are approximated with partial KV, however, we still get the full attention scores for all queries. In this paper, it seems that you only select a subset of queries, does this means some of the queries are pruned? Does this means it is somehow similar to the previous work on token pruning?

---

> ### Author Response · Authors · 2025-11-17
>
> The authors would first like to thank reviewer gbPS for taking the time to read and review our paper, and for their insightful questions. We would like to address these one by one:
>
> •	Your question about chunked prefill is very important. Simply put, chunked prefill is not required for our method and it would work without chunked prefill. Conversely, we believe it’s one of the only methods designed to work specifically with chunked prefill. I’m afraid we did not make this as clear as we would have liked in the original paper, but will try to emphasize this in the final manuscript.
>
> •	Your confusion is entirely justified on this point, we should have emphasized this more. In essence the point we were trying to make was that sparse attention approaches relying on custom kernels have more overhead due to the required deployment of a dynamic compute graph (which can be slower on some hardware) and the fact that, even if the attention computation is sparse, generally all keys and values must be moved onto whatever device is doing that computation. This can be costly from a memory bandwidth standpoint, particularly when it is done repeatedly as in the case of chunked prefill and generation.
>
> •	Again, I think your confusion makes sense, we will try to clean up the explanation of this point in the final manuscript. Put simply, the point we were trying to make was that due to the recursive nature of chunked prefill small errors computed in the initial steps (due to kv-cache subselection in this case) tend to compound throughout the prefill process. If attention is only computed once (as in the non-chunked-prefill case, this compounding doesn’t occur.)
>
> •	Thank you for pointing this out! We will fix it.
>
> •	I agree the term “portability” is vague. What we meant was that many sparse attention methods are designed for particular hardware (through custom kernels) or particular models and that we endeavored to design a method that was both hardware and model agnostic.
>
> •	I think this is probably the most critical point of the paper, so I’m sorry it wasn’t made more clearly. The essence of the problem is this: we would like to assign to each KV pair a score during prefill, which can be used to subselect the KV pairs for the attention computation.  This is often done using attention scores which have shape: (batch size, number of heads Q, number of query tokens, number of key tokens). The score must be of shape  (batch size, number of heads K, number of key tokens) where generally, since most modern models use GQA, we have that number of heads K < number of heads Q. What that means is that once we have some kind of score, we need to aggregate across the queries being considered. We observed that many of the queries interact similarly with the keys, and hence contained somewhat redundant information. Removing some of those queries from the score calculation makes this computation more efficient. Thus, we only need a subset of those queries to compute the score that then is aggregated and used to subselect KVs which then can be used in the attention computation. However, to be clear, we are not subselecting queries in the final attention calculation, just for the computation of the score. The number of queries in (softmax(QK^T)V) after KV subselection remains the same.
>
> Once again, thank you for your thoughtful review, we appreciate your feedback. If you have further questions or concerns we would be more than happy to address them.

---

### Meta-Review · Area_Chair_WJ6M · 2025-12-02

**Summary:**

This paper proposes QUOKA, a training-free sparse attention method designed to improve chunked prefill efficiency in LLMs. It identifies a subset of "critical" queries, those with low cosine similarity to the mean, and uses them to compute importance scores that drive KV-pair subselection. The method is evaluated across several SOTA open-source LLMs and long-context benchmarks, showing consistent speedups and competitive accuracy.

Across the four reviews, the paper was generally well-received for clarity, experimental breadth, and practical relevance, with two reviewers recommending acceptance (8, 6), one on the borderline positive side (6), and one borderline negative (4). The primary concerns were conceptual motivation behind the query-selection heuristic, efficiency in the generation phase, missing baselines, and hardware execution details. The rebuttal addressed many of these issues convincingly.

**Reviewer Concerns:**

Addressed:
* Relationship to chunked prefill and whether it is required.
* Clarification of the query subselection mechanism.
* Motivation for selecting outlier queries, with additional empirical evidence.
* Missing baselines.
* Generation-phase efficiency.
* Hardware considerations for small chunk sizes and discontinuous KV.

Partially unresolved:
* The heuristic still lacks strong theoretical grounding.
* Comparison to clustering-based selection remains limited.
* Fairness of hardware benchmarking (dense baseline may be disadvantaged by small chunks).

**Reviewer Scores:**

* Reviewer gbPS (8): unchanged.
* Reviewer gdtt (6): unchanged or +2.
* Reviewer nDFH (4): likely +2.
* Reviewer JhZf (6): unchanged or +2.

---

### Decision · Program_Chairs · 2026-01-26

Accept (Poster)